# Logical Guidance for the Exact Composition of Diffusion Models

**Francesco Alesiani** [* 1]   **Jonathan Warrell** [* 2]   **Tanja Bien** [* 3]
**Henrik Christiansen** [1]   **Matheus Ferraz** [4]   **Mathias Niepert** [1 3]

## Abstract

We propose LOGDIFF (**Lo**gical **G**uidance for the Exact Composition of **Diff**usion Models), a guidance framework for diffusion models that enables principled constrained generation with complex logical expressions at inference time. We study when exact score-based guidance for complex logical formulas can be obtained from guidance signals associated with atomic properties. First, we derive an exact Boolean calculus that provides a sufficient condition for exact logical guidance. Specifically, if a formula admits a circuit representation in which conjunctions combine conditionally independent subformulas and disjunctions combine subformulas that are either conditionally independent or mutually exclusive, exact logical guidance is achievable. In this case, the guidance signal can be computed exactly from atomic scores and posterior probabilities using an efficient recursive algorithm. Moreover, we show that, for commonly encountered classes of distributions, any desired Boolean formula is compilable into such a circuit representation. Second, by combining atomic guidance scores with posterior probability estimates, we introduce a hybrid guidance approach that bridges classifier-guidance and classifier-free guidance, applicable to both compositional logical guidance and standard conditional generation. We demonstrate the effectiveness of our framework on multiple image and protein structure generation tasks.[1]

## 1. Introduction

Diffusion models have achieved remarkable success in generating high-fidelity data across diverse modalities, from image and video generation (Du et al., 2023; Liu et al., 2021; 2022; Zhu et al., 2024) to protein design (Yang et al., 2024; Abramson et al., 2024) and planning (Ajay et al., 2023; Janner et al., 2022). A key property of these models is their steerability, which allows controlling outputs at inference time via guidance (Ho & Salimans, 2021; Dhariwal & Nichol, 2021; Bansal et al., 2023). This capability extends to compositional generation, where complex guidance terms are composed from combinations of individual concepts (Hinton, 2002; Liu et al., 2022; Du et al., 2023).

Existing compositional guidance methods, however, remain limited. Most approaches combine conditions by heuristically averaging conditional outputs (Liu et al., 2022). While effective for simple conjunctions, such heuristics fail to capture the structure of general logical reasoning, and in particular do not extend naturally to disjunctions, negations, or more complex Boolean expressions. Other recent approaches consider the superposition of distributions (Skreta et al., 2025b), but do not provide a general framework for compositional reasoning. In concurrent work, (Blohm & Garg, 2026) provide a fuzzy-logic-based framework for composing distributions; while this work allows complex Boolean expressions to be represented, Boolean identities are only approximately satisfied, and the framework does not directly ensure compositionality of properties at terminal time.

To address this issue, we introduce LOGDIFF, ***Lo**gical **G**uidance for the Exact Composition of **Diff**usion Models*, a framework connecting Boolean logic and compositional diffusion. We formalize logical constraints as probabilistic events, deriving an exact Boolean calculus in which the combination of conditional outputs is dynamic and depends on the time-varying probability of clauses, rather than on constant weights, as shown in Figure 1. We derive constructible, recursive guidance rules that implement this calculus using only standard diffusion outputs and posterior probability estimators. We provide a detailed example using our recursive guidance rules in Appendix B.

Our main contributions are as follows:

---

[*]Equal contribution  [1]NEC Laboratories Europe  [2]NEC Laboratories America  [3]University of Stuttgart  [4]NEC OncoImmunity.  Correspondence to:  Francesco Alesiani <francesco.alesiani@neclab.eu>,  Jonathan Warrell <jwarrell@nec-labs.com>,  Tanja Bien <tanja.bien@ki.uni-stuttgart.de>, Mathias Niepert <mathias.niepert@neclab.eu>.

*Proceedings of the 43rd International Conference on Machine Learning*, Seoul, South Korea. PMLR 306, 2026. Copyright 2026 by the author(s).

[1]Code is available at: https://github.com/TanjaBien/LogDiff

*Figure 1.* **Logical Compositional Guidance.** Visualization of logical composition using logical scores $s_t(\varphi, \boldsymbol{x})$ for two specific queries $\varphi$. Our framework replaces constant mixing weights with probability-dependent coefficients derived explicitly from posterior probabilities, allowing for mathematically grounded composition.

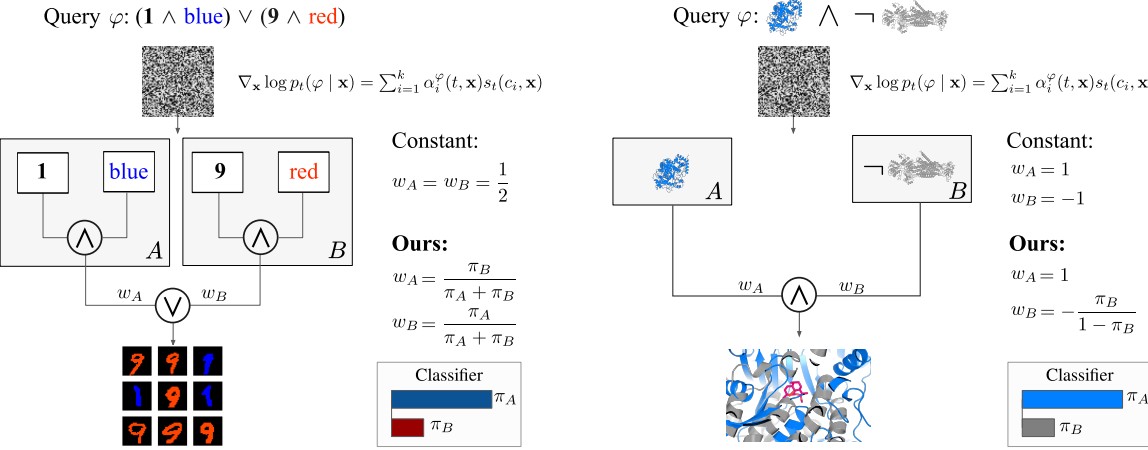

- **Exact Boolean calculus for composition.** We derive an exact calculus for composing models defined by Boolean formulas over atomic predicates. We provide sufficient conditions on the circuit representation of Boolean formulas for our calculus to apply, and show that for certain commonly encountered classes of distribution, any desired Boolean formula is compilable into such a circuit representation.

- **Constructible logical guidance.** We derive practical guidance rules that realize Boolean operators using standard (conditional) diffusion outputs and posterior likelihood scalars, extending classifier-free guidance to logical composition.

- **Hybrid classifier-assisted guidance.** We propose an efficient guidance strategy that combines standard classifier-free guidance with posterior probability estimates to compute the posterior conditioning term.

## 2. Preliminaries

To develop our framework, we build on Classifier-free diffusion guidance (Ho & Salimans, 2021) and Boolean compositionality (Brown, 2003).

### 2.1. Classifier-Free Guidance of Diffusion Models

Diffusion models describe the generative process as two Stochastic Differential Equations (SDEs): the forward process during training and the reverse process at inference. We denote by $\boldsymbol{X}_t \in \mathcal{X} \subseteq \mathbb{R}^d$ the (random) state of the *reverse-time* diffusion at time $t \in [0, T]$. Boldface $\boldsymbol{X} = (\boldsymbol{X}_t)_{t \in [0,T]}$ denotes the entire stochastic process. Lowercase $\boldsymbol{x}_t$ denotes realizations. We treat $t = 0$ as the terminal time, and the sampler integrates the reverse-time SDE from $t = T$ down

to $t = 0$. The reverse-time SDE used for generation is

$$\mathrm{d}\boldsymbol{X}_t = b_t(\boldsymbol{X}_t)\,\mathrm{d}t + \sigma_t(\boldsymbol{X}_t)\,\mathrm{d}\boldsymbol{W}_t, \qquad t \in [0, T], \quad (1)$$

where $\boldsymbol{W}_t$ is a standard $d$-dimensional Wiener process, $b_t : \mathcal{X} \to \mathbb{R}^d$ is the drift and $\sigma_t : \mathcal{X} \to \mathbb{R}^{d \times d}$ the diffusion coefficient. We write $a_t(\boldsymbol{x}) := \sigma_t(\boldsymbol{x})\sigma_t(\boldsymbol{x})^\top$ for the diffusion matrix and $p_t(\boldsymbol{x})$ for the marginal density of $\boldsymbol{X}_t$. The reverse-time drift may be expressed as

$$b_t(\boldsymbol{x}) = f_t(\boldsymbol{x}) - a_t(\boldsymbol{x})\nabla_{\boldsymbol{x}} \log p_t(\boldsymbol{x}), \quad (2)$$

where $f_t$ is the drift term of the forward diffusion process. We refer to $\nabla_{\boldsymbol{x}} \log p_t(\boldsymbol{x})$ as the unconditional score.

We recall the principle of *classifier-free guidance* (CFG) that underlies most conditional diffusion samplers. Let $p_t(\boldsymbol{x})$ denote the unconditional diffusion marginal at time $t$ and $p_t(\boldsymbol{x} \mid c)$ the conditional marginal corresponding to a condition or attribute $c$. By Bayes' rule,

$$p_t(\boldsymbol{x} \mid c) \propto p_t(c \mid \boldsymbol{x})\,p_t(\boldsymbol{x}), \qquad \Rightarrow$$
$$\nabla_{\boldsymbol{x}} \log p_t(\boldsymbol{x} \mid c) = \nabla_{\boldsymbol{x}} \log p_t(\boldsymbol{x}) + \nabla_{\boldsymbol{x}} \log p_t(c \mid \boldsymbol{x}). \quad (3)$$

Hence, the conditional score decomposes into two terms: the unconditional score $\nabla_{\boldsymbol{x}} \log p_t(\boldsymbol{x})$ and the *posterior score* $\nabla_{\boldsymbol{x}} \log p_t(c \mid \boldsymbol{x})$.

Classifier-free guidance uses a single diffusion model trained both unconditionally and conditionally, providing estimates of $\nabla_{\boldsymbol{x}} \log p_t(\boldsymbol{x})$ and $\nabla_{\boldsymbol{x}} \log p_t(\boldsymbol{x} \mid c)$. Interpolating between these scores yields the guided score

$$\nabla_{\boldsymbol{x}} \log p_t(\boldsymbol{x}) + w\big(\nabla_{\boldsymbol{x}} \log p_t(\boldsymbol{x} \mid c) - \nabla_{\boldsymbol{x}} \log p_t(\boldsymbol{x})\big), \quad (4)$$

where $w \geq 0$ controls the conditioning strength. For $w = 1$ this recovers the exact conditional score $\nabla_{\boldsymbol{x}} \log p_t(\boldsymbol{x} \mid c)$, while $w > 1$ amplifies the effect of conditioning.

*Table 1.* Recursive guidance rules for posterior probabilities and scores, which can be implemented as a recursive algorithm (Algorithm 2).

| Expression $\varphi$ | Posterior Probability $\widehat{\pi}(\varphi)$ | Score $s_t(\varphi, \boldsymbol{x})$ |
| --- | --- | --- |
| Atom $c \in \mathcal{C}$ | $p_t(c \mid \boldsymbol{x})$ | $\nabla_{\boldsymbol{x}} \log p_t(\boldsymbol{x} \mid c) - \nabla_{\boldsymbol{x}} \log p_t(\boldsymbol{x})$ |
| Negation $\neg\psi$ | $1 - \widehat{\pi}(\psi)$ | $-\dfrac{\widehat{\pi}(\psi)}{1 - \widehat{\pi}(\psi)} s_t(\psi, \boldsymbol{x})$ |
| Conjunction $\psi \wedge \chi$ | $\widehat{\pi}(\psi)\widehat{\pi}(\chi)$ | $s_t(\psi, \boldsymbol{x}) + s_t(\chi, \boldsymbol{x})$ |
| Disjunction (OR-CI) $\psi \vee \chi$ | $\widehat{\pi}(\psi) + \widehat{\pi}(\chi) - \widehat{\pi}(\psi)\widehat{\pi}(\chi)$ | $\dfrac{\widehat{\pi}(\psi)\widehat{\pi}(\neg\chi)s_t(\psi, \boldsymbol{x}) + \widehat{\pi}(\chi)\widehat{\pi}(\neg\psi)s_t(\chi, \boldsymbol{x})}{\widehat{\pi}(\psi) + \widehat{\pi}(\chi) - \widehat{\pi}(\psi)\widehat{\pi}(\chi)}$ |
| Disjunction (OR-ME) $\psi \vee \chi$ | $\widehat{\pi}(\psi) + \widehat{\pi}(\chi)$ | $\dfrac{\widehat{\pi}(\psi)s_t(\psi, \boldsymbol{x}) + \widehat{\pi}(\chi)s_t(\chi, \boldsymbol{x})}{\widehat{\pi}(\psi) + \widehat{\pi}(\chi)}$ |

## 2.2. Boolean Compositionality

**Atoms and formulas.** Let $\mathcal{C} = \{c_1, \ldots, c_n\}$ be atomic predicates. Each $c_i$ induces an event $A_i \subseteq \mathcal{X}$ on terminal states (time 0). Boolean formulas $\varphi$ are generated from $\mathcal{C}$ by $\wedge, \vee, \neg$, representing AND, OR and NOT operators respectively, and $\top, \bot$, representing true and false respectively; their semantics are sets $[\![\varphi]\!] \subseteq \mathcal{X}$ with $x \models \varphi \iff x \in [\![\varphi]\!]$. We note that $[\![\top]\!] = \mathcal{X}$ and $[\![\bot]\!] = \emptyset$.

**Backward truth probability and logical score.** For $t \in [0, T]$ and $\boldsymbol{x} \in \mathcal{X}$ define

$$\begin{aligned} p_t(\varphi \mid \boldsymbol{x}) &:= \mathbb{P}(\boldsymbol{X}_0 \in [\![\varphi]\!] \mid \boldsymbol{X}_t = \boldsymbol{x}), \\ s_t(\varphi, \boldsymbol{x}) &:= \nabla_{\boldsymbol{x}} \log p_t(\varphi \mid \boldsymbol{x}). \end{aligned} \quad (5)$$

Let $p_t(\boldsymbol{x} \mid \varphi)$ denote the marginal density of $\boldsymbol{X}_t$ under the process conditioned on the terminal event $\{\boldsymbol{X}_0 \in [\![\varphi]\!]\}$. Following the above, conditioning on $\varphi$ corresponds to replacing the unconditional score $\nabla_{\boldsymbol{x}} \log p_t(\boldsymbol{x})$ in (2) by the conditional score $\nabla_{\boldsymbol{x}} \log p_t(\boldsymbol{x} \mid \varphi)$,

$$b_t^\varphi(\boldsymbol{x}) = f_t(\boldsymbol{x}) - a_t(\boldsymbol{x}) \nabla_{\boldsymbol{x}} \log p_t(\boldsymbol{x} \mid \varphi). \quad (6)$$

Now, by Bayes' rule,

$$b_t^\varphi(\boldsymbol{x}) = f_t(\boldsymbol{x}) - a_t(\boldsymbol{x})\big(\nabla_{\boldsymbol{x}} \log p_t(\boldsymbol{x}) + s_t(\varphi, \boldsymbol{x})\big). \quad (7)$$

## 3. Logical Guidance Framework

We explore the problem of composing guidance scores for complex logical expressions. Let $\mathcal{C} = \{c_1, \ldots, c_N\}$ be a finite collection of Boolean predicates, and let $\varphi$ be a Boolean formula over $\mathcal{C}$, constructed using $\wedge, \vee$, and $\neg$. Our objective is to understand when the score $s_t(\varphi, \boldsymbol{x})$ can be evaluated *exactly* by composing the scores of the atomic events. Exact compositionality depends on the interaction between the logical structure of $\varphi$ and the factorization properties of the joint distribution of the atomic events induced

by the diffusion posterior at $(t, \boldsymbol{x})$. In general, Boolean formulas do not permit exact composition.

### 3.1. Compositional Calculus

The atomic posterior logical scores $s_t(c_i, \boldsymbol{x})$ are available from conditional and unconditional diffusion networks. We will show that an exact composition is achievable whenever the structure of a formula $\varphi$ admits a circuit representation in which

- every conjunction node $\psi \wedge \chi$ combines conditionally independent subformulas (decomposability),

- every disjunction node $\psi \vee \chi$ combines subformulas that are either conditionally independent (OR-CI) or mutually exclusive (OR-ME), so that either the independent-disjunction rule or the mutually-exclusive-disjunction rule applies.

These conditions mirror the semantics of probabilistic circuits with decomposable product nodes and either decomposable or deterministic sum nodes (Darwiche, 2022; Vergari et al., 2021). Under these structural assumptions, each formula $\varphi$ is associated with two recursively computed quantities: a posterior $\widehat{\pi}(\varphi) := p_t(\varphi \mid \boldsymbol{x})$ and a logical score $s_t(\varphi, \boldsymbol{x}) := \nabla_x \log p_t(\varphi \mid \boldsymbol{x})$. The recursive rules are provided in Table 1. For each disjunction node $\psi \vee \chi$, we select the OR-CI rules if the terminal events of $\psi$ and $\chi$ are conditionally independent given $\boldsymbol{X}_t = \boldsymbol{x}$, and the OR-ME rule if they are mutually exclusive. The following proposition summarizes the above (proof provided in Appendix A):

**Proposition 3.1.** *Let $\varphi$ be a propositional formula over atoms $\{c_i\}$. Suppose that $\varphi$ admits a circuit representation whose internal nodes are $\wedge$, $\vee$, and $\neg$, and whose $\wedge$- and $\vee$-nodes satisfy, for every $t \in (0, T]$ and every $\boldsymbol{x} \in \mathcal{X}$:*

1. *For every conjunction node $\psi \wedge \chi$,*

$$p_t(\psi \wedge \chi \mid \boldsymbol{x}) = p_t(\psi \mid \boldsymbol{x})\, p_t(\chi \mid \boldsymbol{x}).$$

2. *For every disjunction node $\psi \vee \chi$, either*

$$p_t(\psi \wedge \chi \mid \boldsymbol{x}) = p_t(\psi \mid \boldsymbol{x})\, p_t(\chi \mid \boldsymbol{x}) \quad \text{(OR-CI)},$$

*or*

$$p_t(\psi \wedge \chi \mid \boldsymbol{x}) = 0 \quad \text{(OR-ME)}.$$

*Assume furthermore that $\widehat{\pi}(\psi) = p_t(\psi \mid \boldsymbol{x})$ for all subformulas $\psi$ of $\varphi$, for every $t \in (0, T]$ and every $\boldsymbol{x} \in \mathcal{X}$, and that for every subformula $\psi$ appearing in the circuit, the map $\boldsymbol{x} \mapsto p_t(\psi \mid \boldsymbol{x})$ is differentiable and $0 < p_t(\psi \mid \boldsymbol{x}) < 1$ for all $t \in (0, T]$, $\boldsymbol{x} \in \mathcal{X}$. Then the recursive rules reproduce exactly the true posterior and logical score for $\varphi$ for every $t \in (0, T]$ and $\boldsymbol{x} \in \mathcal{X}$:*

$$\widehat{\pi}(\varphi) = p_t(\varphi \mid \boldsymbol{x}), \qquad s_t(\varphi, \boldsymbol{x}) = \nabla_{\boldsymbol{x}} \log p_t(\varphi \mid \boldsymbol{x}).$$

We note that, for any propositional formula $\varphi$ admitting a circuit representation satisfying the sufficient conditions of Proposition 3.1, it follows that the corresponding logical score can be expressed as a linear combination of the atomic logical scores, with coefficients that are functions of the posterior estimates $\widehat{\pi}(\cdot)$ (and hence of the atomic posteriors). In particular, there exist scalar functions $\alpha_i^{\varphi}(t, \boldsymbol{x})$ such that

$$\nabla_{\boldsymbol{x}} \log p_t(\varphi \mid \boldsymbol{x}) = \sum_{i=1}^{k} \alpha_i^{\varphi}(t, \boldsymbol{x}) s_t(c_i, \boldsymbol{x}), \qquad (8)$$

where $k$ is the number of atoms appearing in $\varphi$ and each coefficient $\alpha_i^{\varphi}(t, \boldsymbol{x})$ is determined by recursive application of the rules in Table 1.

When a formula $\varphi$ admits a circuit representation satisfying Proposition 3.1, exact logical guidance reduces to a single evaluation of that circuit at each diffusion step. Each atomic predicate $c_i$ requires exactly one posterior $\pi(c_i)$ and one score $s_t(c_i, \boldsymbol{x})$ evaluation. As a result, no additional diffusion model evaluations are needed for composite formulas, and the cost of evaluating $s_t(\varphi, \boldsymbol{x})$ scales linearly with the circuit's size. This mirrors tractability properties of probabilistic circuits with decomposable and deterministic structure.

In several situations, guidance properties can be expressed through a set of conditionally independent categorical variables. Examples include combinations of discrete attributes, such as color, shape, or residue type at multiple positions within a protein. Users often wish to specify constraints directly in terms of which joint configurations of the categorical variables are allowed. For instance, a user may allow only a small set of valid attribute combinations, or

forbid specific combinations while leaving all others unconstrained. The following proposition shows that such queries (*categorical-CI queries*) can be evaluated exactly under our logical guidance framework, provided the categorical variables are conditionally independent (proof provided in Appendix A).

**Proposition 3.2.** *Fix $t \in (0, T]$ and $\boldsymbol{x} \in \mathcal{X}$. Let $Z_1, \ldots, Z_M$ be conditionally independent categorical variables given $\boldsymbol{X}_t = \boldsymbol{x}$, with finite domains $\Omega_1, \ldots, \Omega_M$. For each $m \in [M]$ and $v \in \Omega_m$, define the Boolean predicate $c_{m,v} := (Z_m = v)$. Let $\mathcal{A} \subseteq \Omega_1 \times \cdots \times \Omega_M$ be a set of joint assignments and define*

$$\varphi_{\mathcal{A}} := \bigvee_{(v_1, \ldots, v_M) \in \mathcal{A}} \bigwedge_{m=1}^{M} c_{m, v_m}.$$

*Then the circuit of $\varphi_{\mathcal{A}}$ satisfies the sufficient conditions of Proposition 3.1: every $\wedge$-node is conditionally independent and every $\vee$-node is mutually exclusive. Consequently, the recursive rules compute $p_t(\varphi_{\mathcal{A}} \mid \boldsymbol{x})$ and $s_t(\varphi_{\mathcal{A}}, \boldsymbol{x})$ exactly.*

In addition to such categorical-CI queries, we show in Appendix C that our framework handles exactly a large class of *taxonomy queries*, which relate to properties expressible in a tree-structured taxonomy (Proposition C.1). Further, we show in Appendix C that such large classes of queries may be used to demonstrate that our logical guidance framework is complete for all Boolean queries in distributions over predicates with the properties stated. Particularly, we introduce the notion of *compilability* of a Boolean formula $\varphi$, where $\varphi$ is *compilable* iff we may find a probabilistic circuit evaluable using our logical guidance rules corresponding to a formula $\varphi'$, possibly different from $\varphi$, where the events specified by $\varphi$ and $\varphi'$ are identical in the class of distributions specified. We show in Proposition C.2 that our logical guidance framework is complete in this sense (any Boolean formula is compilable) for distributions consisting of (1) a collection of conditionally independent categorical variables and (2) properties expressible in a taxonomy. We note that (1) includes as a special case the case where the properties are a set of events, all of which are CI. Further, we show that for (2), such completeness of compilability is guaranteed at all time points if it holds for the terminal distribution. Finally, we discuss the case of discrete-time updates in Appendix C.3, and show that an analogue of Proposition 3.1 can be derived (Proposition C.3).

### 3.2. Hybrid Conditional Guidance

The exact rules in Table 1 assume access to the posterior probabilities $p_t(c_i \mid \boldsymbol{x})$ for all atomic predicates $c_i$, which in practice are not directly available. We therefore introduce estimators $\widehat{\pi}(c_i) \approx p_t(c_i \mid \boldsymbol{x})$ for the atomic predicates, and use them to compute the coefficients $\widehat{\pi}$ for the composition

*Table 2.* **Conformity score (%, ↑) on CMNIST and Shapes3D.** Evaluation of atomic, binary, and complex queries, where $N = 2 \ldots 5$ denotes random formulas with $N$ AND/OR operators.

| Method | AND | NOT | OR-ME | OR-CI | $N=2$ | $N=3$ | $N=4$ | $N=5$ |
|---|---|---|---|---|---|---|---|---|
| **CMNIST** | | | | | | | | |
| Unconditional | 0.9 | 90.0 | 19.2 | 19.1 | 28.3 | 35.1 | 38.8 | 48.9 |
| Skreta et al. (2025b) | 26.2 | - | 96.84 | 97.54 | - | - | - | - |
| Liu et al. (2022) | **80.4** | **99.5** | - | - | - | - | - | - |
| LoGDiff | **80.4** | 96.5 | **98.0** | 97.2 | **93.8** | **93.3** | **94.2** | **94.4** |
| **Shapes3D** | | | | | | | | |
| Unconditional | 1.3 | 87.5 | 21.9 | 23.4 | 21.5 | 25.6 | 26.7 | 33.2 |
| Skreta et al. (2025b) | 30.1 | - | 74.4 | 77.1 | - | - | - | - |
| Liu et al. (2022) | **82.4** | **98.7** | - | - | - | - | - | - |
| LoGDiff | **82.4** | 96.4 | **94.3** | **96.0** | **88.8** | **88.6** | **85.1** | **87.6** |

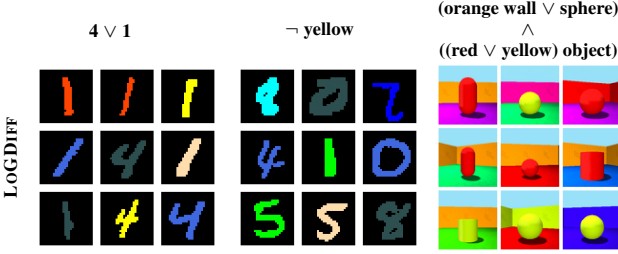

*Figure 2.* **Qualitative results on synthetic datasets.** Samples generated by LoGDiff on CMNIST and Shapes3D for representative logical queries. The examples show that LoGDiff produces diverse valid samples for disjunctions, negations, and recursive queries. For negated queries, e.g., ¬ yellow, the query specifies which attribute value to avoid rather than a single target value to generate.

rules. For example, $\widehat{\pi}(c_i)$ can be obtained by training a noise-aware classifier on $\boldsymbol{x}_t$ across diffusion times $t$.

**Gradient-free guidance.** This provides a hybrid guidance mechanism that separates direction from weighting. The scores are obtained from standard conditional and unconditional diffusion networks (Equation (3)) , while the mixture weights required to compose complex formulas are provided by $\widehat{\pi}(\cdot)$ (Equation (8)). Importantly, this does not require backpropagating through a classifier to obtain gradients: probability estimates enter only as scalar coefficients in closed-form composition rules, while all score gradients are supplied by the diffusion model itself. Hence, our approach bridges classifier-guidance and classifier-free guidance: it preserves the stability and the efficiency of classifier-free score estimation while enabling probabilistically-correct composition for disjunctions and negations.

Moreover, when $\widehat{\pi}(c_i)$ is estimated directly from the diffusion model, e.g., with score-based probability estimators (Li et al., 2023; Skreta et al., 2025b; Koulischer et al., 2026; Karczewski et al., 2025), the resulting method requires only a trained conditional diffusion model with the atomic scores available under classifier-free guidance.

**Adaptive repulsive guidance.** Beyond compositional queries, LoGDiff can also improve standard single-condition generation. When conditioning on a single class $A$, a common failure mode of diffusion models is confusion with other similar classes. Our framework naturally supports adaptive *repulsive guiding* by conditioning on formulas of the form $A \wedge \neg B$, where $B$ denotes a competing class. The resulting guidance direction is a probability-weighted combination of the atomic scores for $A$ and $B$, with weights determined by $\widehat{\pi}(A \mid \boldsymbol{x})$ and $\widehat{\pi}(B \mid \boldsymbol{x})$, unlike heuristic approaches that rely on constant weights (Shenoy et al., 2024). As a result, repulsive guidance is applied strongly only in regions where $B$ is locally probable, and vanishes when $B$ is already unlikely. An example is shown in Appendix B for illustration.

## 4. Experiments

We evaluate LoGDiff on image and molecular generation tasks to assess both logical controllability, generation quality, and generality across modalities. For image generation, we study whether LoGDiff satisfies increasingly complex logical queries and preserves sample quality under composition. We then analyze repulsive guiding as an inference-time mechanism for improving sample quality. Finally, we apply LoGDiff to structure-based drug design, demonstrating that the proposed guidance rules generalize to protein-conditioned molecular generation and multi-target objectives.

### 4.1. Image generation

To evaluate our method, we employ three distinct experimental settings: 1) synthetic datasets (CMNIST, Shapes3D) to establish the model's ability to satisfy complex logical queries; 2) CelebA to ensure that our guidance mechanism preserves image quality in real-world domains; and 3) ImageNet and the synthetic datasets to analyze the impact of repulsive guiding on sample quality.

**Datasets and metrics.** To evaluate logical guidance, we utilize Colored MNIST (CMNIST) (LeCun et al., 1998; Gaudi et al., 2025) and Shapes3D (Kim & Mnih, 2018). We quantify performance using the Conformity Score (CS) (Gaudi et al., 2025), defined as the percentage of generated samples that satisfy the target logical formula according to a pre-trained classifier. To ensure that high conformity does not come at the cost of reduced diversity, we report the Mean Batch Joint Entropy $H$. For real-world attributes on CelebA (Liu et al., 2015), we additionally report Fréchet Inception Distance (FID) to ensure visual fidelity is maintained. For more detail see Appendix E.2.

**Baselines.** We compare our method against the composition operators proposed by Liu et al. (2022) and Skreta et al. (2025b). Since these approaches do not define recursive

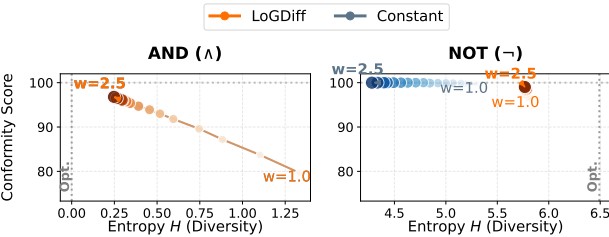

*Figure 3.* **Conformity-diversity trade-off on CMNIST**. Conformity Score (↑) vs. Joint Shannon Entropy across varying guidance scales $w \in [1.0, 2.5]$. Vertical dotted lines indicate the theoretical optimal entropy for each task (note that for AND, the optimal entropy is low as the solution space is highly constrained). While the constant baseline from Liu et al. (2022) (blue) suffers from low entropy, indicating mode collapse, as guidance strength increases, our method (orange) successfully maintains high sample diversity while achieving high conformity scores.

composition rules for arbitrary logical expressions, we restrict the comparison to the operations explicitly defined in their respective frameworks. An overview of the corresponding composition rules is provided in Table 20.

**Recursive task complexity.** We evaluate the methods through logical queries of increasing complexity, quantified by the count $N$ of AND/OR operators. The evaluation ranges from single-operator baselines to nested formulas with up to $N = 5$ logical operators. Queries are generated randomly while enforcing logical validity (e.g., avoiding impossible intersections like $3 \wedge 4$, see Appendix E.1).

**Results on synthetic datasets.** As detailed in Table 2, LoGDIFF consistently outperforms Skreta et al. (2025b) across all operations for which this baseline is defined. For conjunctions, LoGDIFF coincides with Liu et al. (2022), while Liu et al. (2022) performs slightly better on negation on the synthetic datasets. Since the baselines do not define composition rules for recursive logical queries, we compare them only on the unary and binary operations they support. In contrast, LoGDIFF extends to complex recursive queries and remains stable under increasing compositional complexity, maintaining high conformity scores for $N = 2 \ldots 5$. Figure 2 shows qualitative examples for logical queries of varying complexity. More qualitative results, as well as additional analyses of compute time and the effect of using the classifier versus the estimator, are provided in Appendix G.2. For negated queries, valid samples may take any value of the negated attribute except the excluded one. Unconditional generation results are provided as a lower-bound reference, illustrating the probability of satisfying constraints by chance.

**Results on real-world images.** On CelebA, we evaluate logical composition on two attributes: gender (binary) and hair color (multiattribute). As shown in Table 3, LoGDIFF out-

performs Skreta et al. (2025b). For conjunctions, LoGDIFF matches Liu et al. (2022), while for negation both methods achieve similar CS. However, LoGDIFF attains a substantially lower FID for negation, suggesting that probability-weighted negation avoids the target concept without overly suppressing related visual features. This highlights a limitation of constant-weight negation: while increasing the strength of the negative guidance can yield high CS, it may also reduce diversity or degrade image quality. We further observe this behavior on CMNIST in Figure 3, where constant-weight negation maintains high conformity but exhibits a stronger reduction in entropy as guidance increases. In contrast, LoGDIFF preserves diversity more effectively while maintaining high conformity. For negation and disjunction, LoGDIFF achieves FID close to unconditional generation, indicating that logical composition preserves visual quality.

*Table 3.* Conformity Score (CS) and FID on CelebA.

| Method | AND CS ↑ | AND FID ↓ | NOT CS ↑ | NOT FID ↓ | OR-CI CS ↑ | OR-CI FID ↓ | OR-ME CS ↑ | OR-ME FID ↓ |
|---|---|---|---|---|---|---|---|---|
| Unconditional | 8.3 | **12.2** | 64.6 | 12.2 | 61.2 | 12.2 | 45.0 | **12.2** |
| Skreta et al. (2025b) | 23.3 | 63.2 | - | - | 89.7 | 13.9 | 77.8 | 15.4 |
| Liu et al. (2022) | **61.1** | 26.5 | 92.7 | 19.7 | - | - | - | - |
| LoGDIFF | **61.1** | 26.5 | **94.3** | **12.1** | **94.5** | **11.2** | **84.0** | 12.3 |

**Enhancing generation quality via repulsive guiding.** We investigate whether *repulsive guiding* can enhance the generation quality of single-class conditional generation for ImageNet (with optimal FID settings (Karras et al., 2024)) and compositional guidance for synthetic datasets. Concretely, we replace every atomic condition $A$ with a logical query of the form $A \wedge \neg B$, where $A$ is the desired class and $B$ is a competing class. Rather than fixing $B$ globally, we select it adaptively at each diffusion step based on the current noisy sample $\boldsymbol{x}_t$, choosing the most probable non-$A$ class under a noise-aware classifier $\widehat{\pi}$. We then apply our compositional construction to compute the corresponding logical score using the atomic diffusion scores together with the estimates $\widehat{\pi}(B \mid \boldsymbol{x}_t)$. This results in an attractive term toward $A$ and a repulsive term away from the currently most plausible $B$, but only in regions where $B$ is probable. In this way, our method provides a principled guidance mechanism that empirically improves FID by suppressing class confusions while preserving sample diversity. As shown in Figure 4 and Table 4, our method provides a principled, state-dependent guidance mechanism that empirically improves FID and CS by suppressing class confusions while preserving sample diversity.

### 4.2. Molecular Generation

We explore LoGDIFF for multi-target structure-based drug design, where the objective is to design ligands (i.e., small drug-like molecules) given a target protein in its three-

*Table 4.* Effect of repulsive guidance (RG) on conformity score (%, ↑) on CMNIST and Shapes3D.

| Method | AND | NOT | OR-ME | OR-CI | $N=2$ | $N=3$ | $N=4$ | $N=5$ |
|---|---|---|---|---|---|---|---|---|
| **CMNIST** | | | | | | | | |
| LoGDiff | 80.4 | 96.5 | **98.0** | 97.2 | 93.8 | 93.3 | 94.2 | 94.4 |
| LoGDiff + RG | 83.6 | 98.4 | 97.9 | 98.0 | 94.7 | 94.3 | 95.1 | 95.5 |
| **Shapes3D** | | | | | | | | |
| LoGDiff | 82.4 | 96.4 | 94.3 | 96.0 | 88.8 | 88.6 | 85.1 | 87.6 |
| LoGDiff + RG | 86.4 | 99.3 | 95.6 | 96.7 | 91.8 | 90.5 | 89.8 | 89.9 |

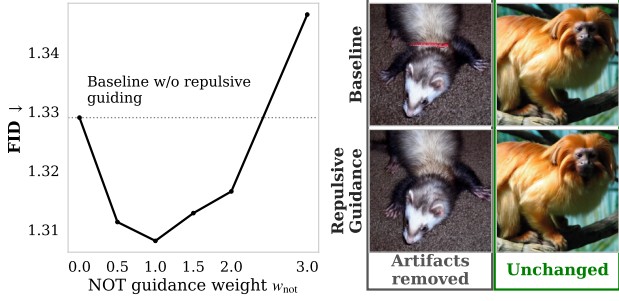

*Figure 4.* **Impact of adaptive repulsive guidance.** (Left) Moderate guidance weights improve FID scores compared to no repulsive guiding ($w_{\text{not}} = 0.0$). (Right) Repulsive guiding removes artifacts while clearly defined samples remain unchanged.

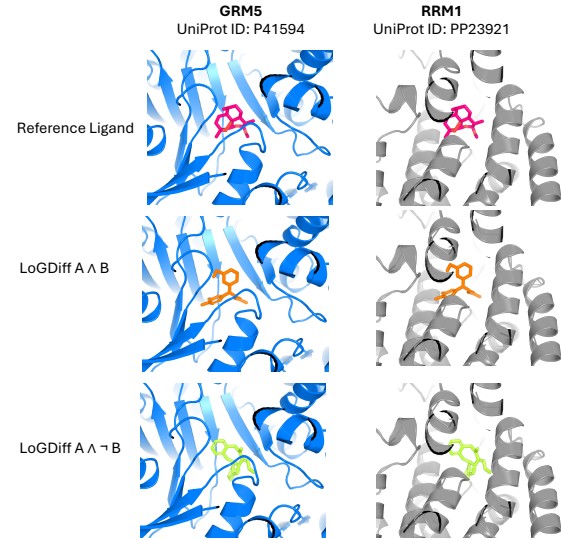

*Figure 5.* **Visualization of ligands in the GRM5-RRM1 dual-target binding site.** (Top) Reference ligand in the aligned binding pocket. (Middle) Representative ligand generated under the guidance term ($A \wedge B$). (Bottom) Representative ligand generated under the selective constraint ($A \wedge \neg B$), designed to engage GRM5 while avoiding RRM1 binding. Protein surfaces are shown for GRM5 (target A) and RRM1 (target B), with the ligand displayed as sticks.

dimensional structure. We experiment with the dual-target drug design, in which the ligand simultaneously binds two proteins. Dual-target drug design is of interest for combating various cancers and neurodegeneration (Ramsay et al., 2018), or in reducing drug resistance (Yang et al., 2024). We study the performance of compositional guidance, where the two atomic conditions are two target proteins, using the experimental setup of (Skreta et al., 2025a). We evaluate our framework on the GRM5-RRM1 protein pair (UniProt: P41594, P23921), where GRM5 is a metabotropic glutamate receptor implicated in neurological disorders, and RRM1 is a ribonucleotide reductase subunit that plays a central role in DNA synthesis and is a validated oncology target. Additional target pairs' evaluations are reported in Appendix F. The generated ligand performance is assessed by the docking score to each target protein using AutoDock Vina (Eberhardt et al., 2021), where 32 ligands of size 23 are generated over 8 experiments.

Tables 5 and 6 list the results of LoGDiff, TargetDiff (Guan et al., 2023), and DualDiff (Yang et al., 2024), where we perform guidance with fixed mixing weights. Tables 7 and 8 show the impact of Feynman-Kac Correction (FKC) (Skreta et al., 2025a). Table 5, similar to (Yang et al., 2024; Skreta et al., 2025a), evaluates the performance for dual targets (AND and OR), while Table 6 lists the results for one on-target and one off-target use case (AND-NOT and XOR). We report the average docking score for each target. A lower docking score indicates better binding.

The impact of logical guidance on ligand geometry is illus-

trated in Figure 5, where $A \wedge B$ ligands occupy the shared binding pocket with complementary interactions to both targets, while $A \wedge \neg B$ ligands adopt distinct poses that favor GRM5 binding while minimizing RRM1 contacts. A detailed 2D interaction analysis illustrating the distinct binding profiles of ligands generated under different logical constraints is provided in Supplementary Figure 6.

We also report the average difference between the minimum and maximum docking scores under guidance. In Table 6, the expected behavior is an increase in the maximum score and a decrease in the minimum score, consistent with improved on-target binding and reduced off-target binding. To the best of our knowledge, this use case has not been systematically studied in prior ligand–protein drug design work. The AND-NOT formulation enables selective target engagement, generating ligands that bind one protein while avoiding another. As shown in Table 6, LoGDiff maintains stable generation compared to other methods. The XOR composition ($A \oplus B$) allows the generative model to autonomously select which target to prioritize, achieving comparable performance.

The validity and uniqueness (V. & U.) of the generated ligands are also presented, as well as diversity and quality metrics summarizing the drug likeness (QED (Bickerton et al., 2012)) and their synthetic accessibility (SA (Ertl & Schuffenhauer, 2009)). Notably, the OR composition (A ∨ B) with tempering yields product scores approaching those of the AND composition while providing greater flexibility

*Table 5.* Evaluation of LOGDIFF to generate drug candidates with dual on-targets, along with the baselines: TargetDiff (Guan et al., 2023), and DualDiff (Yang et al., 2024). A higher average docking score should correlate with a higher binding affinity to both targets.

| | $(A * B)\uparrow$ | $A\downarrow$ | $B\downarrow$ | Div. $\uparrow$ | V. & U. $\uparrow$ | Qual. $\uparrow$ |
|---|---|---|---|---|---|---|
| TargetDiff | $71.01_{\pm2.96}$ | $-8.98_{\pm0.21}$ | $-7.86_{\pm0.15}$ | $0.88_{\pm0.01}$ | $0.94_{\pm0.04}$ | $0.28_{\pm0.04}$ |
| $A \wedge B$ | | | | | | |
| DualDiff | $71.87_{\pm3.33}$ | $-8.85_{\pm0.24}$ | $-8.08_{\pm0.17}$ | $0.89_{\pm0.01}$ | $0.97_{\pm0.03}$ | $0.27_{\pm0.10}$ |
| LOGDIFF | $73.20_{\pm3.18}$ | $-8.99_{\pm0.24}$ | $-8.11_{\pm0.17}$ | $0.89_{\pm0.01}$ | $0.98_{\pm0.02}$ | $0.26_{\pm0.08}$ |
| $A \vee B$ | | | | | | |
| LOGDIFF | $\mathbf{73.91}_{\pm0.71}$ | $-9.01_{\pm0.03}$ | $-8.16_{\pm0.10}$ | $0.90_{\pm0.00}$ | $0.99_{\pm0.01}$ | $0.16_{\pm0.05}$ |

*Table 6.* Evaluation of LOGDIFF with on and off targets, along with the baselines: TargetDiff and DualDiff. The $A \wedge \neg B$ uses $A$ as on-target and $B$ as off-target; in the $A \oplus B$, the roles of on- and off-targets are left to the generative model.

| | $\Delta(A, B)\uparrow$ | $A\downarrow$ | $B\uparrow$ | Div. $\uparrow$ | V. & U. $\uparrow$ | Qual. $\uparrow$ |
|---|---|---|---|---|---|---|
| $A \wedge \neg B$ | | | | | | |
| DualDiff | $0.28_{\pm0.08}$ | $-13.35_{\pm1.07}$ | $-13.53_{\pm1.23}$ | $0.85_{\pm0.01}$ | $0.91_{\pm0.09}$ | $0.00_{\pm0.00}$ |
| SDE | $0.86_{\pm0.13}$ | $-9.06_{\pm0.16}$ | $-8.24_{\pm0.08}$ | $0.89_{\pm0.00}$ | $0.99_{\pm0.03}$ | $0.28_{\pm0.05}$ |
| LOGDIFF | $\mathbf{0.94}_{\pm0.24}$ | $-9.01_{\pm0.19}$ | $-8.11_{\pm0.14}$ | $0.90_{\pm0.00}$ | $1.00_{\pm0.00}$ | $0.26_{\pm0.11}$ |
| $A \oplus B$ | | | | | | |
| | $\Delta(A, B)\uparrow$ | $A$ | $B$ | Div. $\uparrow$ | V. & U. $\uparrow$ | Qual. $\uparrow$ |
| LOGDIFF | $0.89_{\pm0.13}$ | $-9.00_{\pm0.03}$ | $-8.15_{\pm0.11}$ | $0.90_{\pm0.00}$ | $0.99_{\pm0.01}$ | $0.16_{\pm0.05}$ |

*Table 7.* Evaluation of FKC (Skreta et al., 2025a) when generating drug candidates with dual on-targets. FKC improves performance for the AND case while reducing performance for OR.

| | $(A * B)\uparrow$ | $A\downarrow$ | $B\downarrow$ | Div. $\uparrow$ | V. & U. $\uparrow$ | Qual. $\uparrow$ |
|---|---|---|---|---|---|---|
| $A \wedge B$ **(FKC)** | | | | | | |
| LOGDIFF | $80.89_{\pm6.60}$ | $-9.57_{\pm0.47}$ | $-8.43_{\pm0.34}$ | $0.73_{\pm0.04}$ | $0.77_{\pm0.16}$ | $0.23_{\pm0.20}$ |
| $A \vee B$ **(FKC)** | | | | | | |
| LOGDIFF | $73.51_{\pm5.42}$ | $-9.02_{\pm0.33}$ | $-8.13_{\pm0.30}$ | $0.69_{\pm0.04}$ | $0.84_{\pm0.07}$ | $0.25_{\pm0.11}$ |

*Table 8.* Evaluation of FKC when generating drug candidates with one on-target and one off-target. As with the dual-target case, FKC improves the performance of LOGDIFF, especially for the XOR case.

| | $\Delta(A, B)\uparrow$ | $A\downarrow$ | $B\uparrow$ | Div. $\uparrow$ | V. & U. $\uparrow$ | Qual. $\uparrow$ |
|---|---|---|---|---|---|---|
| $A \wedge \neg B$ **(FKC)** | | | | | | |
| SDE | $0.96_{\pm0.34}$ | $-8.75_{\pm0.32}$ | $-7.79_{\pm0.46}$ | $0.77_{\pm0.08}$ | $0.73_{\pm0.24}$ | $0.30_{\pm0.24}$ |
| LOGDIFF | $0.96_{\pm0.35}$ | $-8.84_{\pm0.84}$ | $-7.92_{\pm0.93}$ | $0.78_{\pm0.07}$ | $0.83_{\pm0.15}$ | $0.25_{\pm0.20}$ |
| $A \oplus B$ **(FKC)** | | | | | | |
| | $\Delta(A, B)\uparrow$ | $A$ | $B$ | Div. $\uparrow$ | V. & U. $\uparrow$ | Qual. $\uparrow$ |
| LOGDIFF | $1.00_{\pm0.57}$ | $-9.52_{\pm0.50}$ | $-8.61_{\pm0.55}$ | $0.69_{\pm0.02}$ | $0.83_{\pm0.04}$ | $0.33_{\pm0.24}$ |

in target engagement. The method preserves molecular quality metrics competitive with specialized dual-target methods like DualDiff. We extract posterior probability estimates from the diffusion model for the logical guidance rules as explained in Appendix D.4.

## 5. Related Work

The composition of diffusion models is typically realized either through their connection to energy-based models (EBMs) (Du et al., 2023; Nie et al., 2021; Ajay et al., 2023) or via linear combinations of learned score functions (Liu et al., 2022; Kong et al., 2025). The latter view can be interpreted as a generalized form of guidance (Ho & Salimans, 2021; Dhariwal & Nichol, 2021; Bansal et al., 2023), in which conditional score estimates are combined using fixed scalar weights. In this work, we adopt this guidance-based perspective but extend it to complex Boolean expressions, while drawing on related work (Li et al., 2023; Skreta et al., 2025b) to replace fixed weights with probability-dependent posterior coefficients.

A parallel line of work studies Boolean compositions of density functions, for example those derived from pre-trained models, to merge their capabilities. This is achieved either through weight averaging (Zhong et al., 2024; Biggs et al., 2024) or via inference-time density composition, including probability-weighted estimators (Skreta et al., 2025b), fuzzy logic operators (Blohm & Garg, 2025), and classifier guidance (Garipov et al., 2023). In contrast to approaches that compose multiple models, we focus on attribute composition using scores derived from a single conditional diffusion model. We note that (Gaudi et al., 2025) also considers logical composition, but address it by modifying training objectives to encourage attribute independence. Their approach complements ours and can be naturally integrated with the proposed logical guidance rules. Further prior work has explored limited forms of distribution composition for molecular systems, including ligand–protein design (Yang et al., 2024), distribution annealing and product-of-experts formulations (Skreta et al., 2025a), combinations of learned distributions (Skreta et al., 2025b), and antibody design (Alesiani et al., 2025).

Finally, we summarize the relationship of our specific operator set (Table 1) to prior and independent work as follows: (a) to our knowledge, our NOT operator has not been investigated in prior work, aside from work developed independently of ours (Koulischer et al., 2025), where it was proposed not as a compositional operator, but in the context of negative guidance; our AND operator has been widely used, for instance in (Gaudi et al., 2025; Skreta et al., 2025a; Blohm & Garg, 2026), although without explicitly stating a CI assumption except for (Gaudi et al., 2025); (c) our OR-CI operator has not to our knowledge been considered in prior work; (d) our OR-ME has been widely used in previous works as disjunction, including (Skreta et al., 2025b; Blohm & Garg, 2026), although without explicitly stating its dependence on an ME assumption. Our approach allows these operators to be unified in a common framework, in which our theoretical analysis demonstrates their sufficiency given particular properties of the distributions they are applied to.

# 6. Discussion

LOGDIFF translates complex logical expressions into exact probabilistically consistent guidance terms: for disjunctions and negations, the guidance depends on the posterior of each formula, rather than fixed coefficients. The sufficient conditions in Proposition 3.1 connect logical guidance to tractable inference in probabilistic circuits. For a formula admitting a circuit representation with decomposable conjunctions and conditionally-independent or mutually-exclusive disjunctions, evaluating $s_t(\varphi, \boldsymbol{x})$ reduces to a linear-time circuit pass at each diffusion step. This connection also clarifies when certain queries can be evaluated exactly, such as those involving constraints defined by mutually independent categorical variables (Proposition 3.2).

**Limitations.** The exactness of the method relies on properties (CI/ME) that may only hold approximately. The method also depends on accurate posterior probability estimates. Deviations of these estimates can be amplified by the nonlinear coefficients, especially for negation through $\widehat{\pi}/(1-\widehat{\pi})$. Practical stabilizations (e.g., clipping) can improve robustness but deviate from the exact theory. Finally, while circuit evaluation is cheap, the overall cost still scales with the number of atomic predicates in the logical expressions.

# Impact Statement

This paper presents work whose goal is to advance the field of Machine Learning. There are many potential societal consequences of our work, none which we feel must be specifically highlighted here.

# Acknowledgements

The authors thank the Deutsche Forschungsgemeinschaft (DFG, German Research Foundation) for supporting this work under Germany's Excellence Strategy–EXC2120/1–390831618. The authors gratefully acknowledge the computing time provided on the high-performance computer HoreKa by the National High-Performance Computing Center at KIT (NHR@KIT). This center is jointly supported by the Federal Ministry of Education and Research and the Ministry of Science, Research and the Arts of Baden-Württemberg, as part of the National High-Performance Computing (NHR) joint funding program (https://www.nhr-verein.de/en/our-partners). HoreKa is partly funded by the German Research Foundation (DFG).

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

## Supplementary Material

## A. Proofs

Proposition 3.1

*Proof.* Fix arbitrary $t \in (0, T]$ and $\boldsymbol{x} \in \mathcal{X}$. We argue by induction over a topological ordering of the circuit nodes.

*Base case (atoms).* For an atomic predicate $c_i$ we have by assumption $\widehat{\pi}(c_i) = p_t(c_i \mid \boldsymbol{x})$ and $s_t(c_i, \boldsymbol{x}) = \nabla_{\boldsymbol{x}} \log p_t(\boldsymbol{x} \mid c_i) - \nabla_{\boldsymbol{x}} \log p_t(\boldsymbol{x})$. Thus the claim holds.

*Inductive step (negation).* Suppose the claim holds for $\psi$. Then by definition, $p_t(\neg\psi \mid \boldsymbol{x}) = 1 - p_t(\psi \mid \boldsymbol{x})$ and

$$
\begin{aligned}
\nabla_{\boldsymbol{x}} \log p_t(\neg\psi \mid \boldsymbol{x}) &= \nabla_{\boldsymbol{x}} \log \left(1 - p_t(\psi \mid \boldsymbol{x})\right) \\
&= -\frac{p_t(\psi \mid \boldsymbol{x})}{1 - p_t(\psi \mid \boldsymbol{x})} \nabla_{\boldsymbol{x}} \log p_t(\psi \mid \boldsymbol{x}),
\end{aligned}
$$

which matches the recursive rule for $s_t(\neg\psi, \boldsymbol{x})$ when $\widehat{\pi}(\psi) = p_t(\psi \mid \boldsymbol{x})$. Likewise $\widehat{\pi}(\neg\psi) = 1 - \widehat{\pi}(\psi) = p_t(\neg\psi \mid \boldsymbol{x})$.

*Inductive step (conjunction).* Let $\varphi = \psi \wedge \chi$ and assume the inductive hypothesis for $\psi$ and $\chi$. By assumption, $p_t(\psi \wedge \chi \mid \boldsymbol{x}) = p_t(\psi \mid \boldsymbol{x})p_t(\chi \mid \boldsymbol{x})$, so with $\widehat{\pi}(\psi) = p_t(\psi \mid \boldsymbol{x})$ and $\widehat{\pi}(\chi) = p_t(\chi \mid \boldsymbol{x})$ we obtain

$$
\widehat{\pi}(\psi \wedge \chi) = \widehat{\pi}(\psi)\widehat{\pi}(\chi) = p_t(\psi \mid \boldsymbol{x})p_t(\chi \mid \boldsymbol{x}) = p_t(\psi \wedge \chi \mid \boldsymbol{x}).
$$

Moreover,

$$
\begin{aligned}
\nabla_{\boldsymbol{x}} \log p_t(\psi \wedge \chi \mid \boldsymbol{x}) &= \nabla_{\boldsymbol{x}} \log p_t(\psi \mid \boldsymbol{x}) + \\
&\quad \nabla_{\boldsymbol{x}} \log p_t(\chi \mid \boldsymbol{x}) \\
&= s_t(\psi, \boldsymbol{x}) + s_t(\chi, \boldsymbol{x}),
\end{aligned}
$$

which is exactly the recursive rule for $s_t(\psi \wedge \chi, \boldsymbol{x})$.

*Inductive step (disjunction).* Let $\varphi = \psi \vee \chi$ and assume the inductive hypothesis for $\psi$ and $\chi$. By inclusion–exclusion,

$$
p_t(\psi \vee \chi \mid \boldsymbol{x}) = p_t(\psi \mid \boldsymbol{x}) + p_t(\chi \mid \boldsymbol{x}) - p_t(\psi \wedge \chi \mid \boldsymbol{x}).
$$

There are two cases.

If $\psi$ and $\chi$ are conditionally independent, then $p_t(\psi \wedge \chi \mid \boldsymbol{x}) = p_t(\psi \mid \boldsymbol{x})p_t(\chi \mid \boldsymbol{x})$ and

$$
\begin{aligned}
\nabla_{\boldsymbol{x}} \log p_t(\psi \vee \chi \mid \boldsymbol{x}) =& \\
\frac{p_t(\psi \mid \boldsymbol{x})\left(1 - p_t(\chi \mid \boldsymbol{x})\right)\nabla_{\boldsymbol{x}} \log p_t(\psi \mid \boldsymbol{x})}{p_t(\psi \mid \boldsymbol{x}) + p_t(\chi \mid \boldsymbol{x}) - p_t(\psi \mid \boldsymbol{x})p_t(\chi \mid \boldsymbol{x})} &+ \\
\frac{p_t(\chi \mid \boldsymbol{x})\left(1 - p_t(\psi \mid \boldsymbol{x})\right)\nabla_{\boldsymbol{x}} \log p_t(\chi \mid \boldsymbol{x})}{p_t(\psi \mid \boldsymbol{x}) + p_t(\chi \mid \boldsymbol{x}) - p_t(\psi \mid \boldsymbol{x})p_t(\chi \mid \boldsymbol{x})}&,
\end{aligned}
$$

which coincides with OR-CI rule for the scores when $\widehat{\pi}(\psi) = p_t(\psi \mid \boldsymbol{x})$ and $\widehat{\pi}(\chi) = p_t(\chi \mid \boldsymbol{x})$. The posterior recursion $\widehat{\pi}(\psi \vee \chi)$ for CI matches $p_t(\psi \vee \chi \mid \boldsymbol{x})$ by the same identity.

If $\psi$ and $\chi$ are mutually exclusive, then $p_t(\psi \wedge \chi \mid \boldsymbol{x}) = 0$ and $p_t(\psi \vee \chi \mid \boldsymbol{x}) = p_t(\psi \mid \boldsymbol{x}) + p_t(\chi \mid \boldsymbol{x})$, and

$$
\begin{aligned}
\nabla_{\boldsymbol{x}} \log p_t(\psi \vee \chi \mid \boldsymbol{x}) &= \frac{p_t(\psi \mid \boldsymbol{x})\nabla_{\boldsymbol{x}} \log p_t(\psi \mid \boldsymbol{x})}{p_t(\psi \mid \boldsymbol{x}) + p_t(\chi \mid \boldsymbol{x})} + \\
&\quad \frac{p_t(\chi \mid \boldsymbol{x})\nabla_{\boldsymbol{x}} \log p_t(\chi \mid \boldsymbol{x})}{p_t(\psi \mid \boldsymbol{x}) + p_t(\chi \mid \boldsymbol{x})},
\end{aligned}
$$

which coincides with $s_t(\psi \vee \chi, \boldsymbol{x})$ for ME under the same identification of $\widehat{\pi}$ with $p_t(\cdot \mid \boldsymbol{x})$. The ME posterior recursion $\widehat{\pi}(\psi \vee \chi) = \widehat{\pi}(\psi) + \widehat{\pi}(\chi)$ matches $p_t(\psi \vee \chi \mid \boldsymbol{x})$.

In all cases, the recursive rules reproduce $p_t(\cdot \mid \boldsymbol{x})$ and $\nabla_{\boldsymbol{x}} \log p_t(\cdot \mid \boldsymbol{x})$ at the node $\varphi$ assuming they do so for its children. By structural induction over the circuit of $\varphi$, the claim holds for all subformulas and in particular for $\varphi$ itself. $\qquad\square$

**Proposition 3.2**

*Proof.* Fix $(t, \boldsymbol{x})$. For any assignment $\alpha = (v_1, \ldots, v_M) \in \mathcal{A}$, define

$$Y_\alpha := \bigwedge_{m=1}^{M} c_{m,v_m}.$$

By conditional independence of $Z_1, \ldots, Z_M$,

$$p_t(Y_\alpha \mid \boldsymbol{x}) = p_t \left( \bigwedge_{m=1}^{M} (Z_m = v_m) \,\middle|\, \boldsymbol{x} \right) \tag{9}$$

$$= \prod_{m=1}^{M} p_t(Z_m = v_m \mid \boldsymbol{x}), \tag{10}$$

so each $\wedge$-node is CI.

If $\alpha \neq \beta$, then there exists $m$ such that $v_m \neq w_m$. Since $Z_m$ is categorical, the events $(Z_m = v_m)$ and $(Z_m = w_m)$ are mutually exclusive, hence $Y_\alpha \wedge Y_\beta = \bot$ and $p_t(Y_\alpha \wedge Y_\beta \mid \boldsymbol{x}) = 0$. Thus, the $\vee$-node is OR-ME.

Therefore, the circuit satisfies the conditions of Prop. 3.1, and the claim follows. $\qquad\square$

## B. Example of Recursive Construction

We illustrate the recursive construction of posterior coefficients and logical scores on the formula from Figure 1

$$\varphi = (1 \wedge \text{blue}) \vee (9 \wedge \text{red}).$$

We assume conditional independence of subformulas and use $\widehat{\pi}_i \approx p_t(c_i \mid \boldsymbol{x})$ and $s_t(c_i, \boldsymbol{x}) = s_i(t, \boldsymbol{x}) - s_\emptyset(t, \boldsymbol{x})$ as atomic input

*(1) Posterior coefficients.*

$$
\begin{aligned}
\widehat{\pi}(1) &= \widehat{\pi}_1, \\
\widehat{\pi}(\text{blue}) &= \widehat{\pi}_{\text{blue}}, \\
\widehat{\pi}(3) &= \widehat{\pi}_9, \\
\widehat{\pi}(\text{red}) &= \widehat{\pi}_{\text{red}}, \\
\widehat{\pi}(\psi) &:= \widehat{\pi}(1 \wedge \text{blue}) = \widehat{\pi}_1 \widehat{\pi}_{\text{blue}} \\
\widehat{\pi}(\chi) &:= \widehat{\pi}(9 \wedge \text{red}) = \widehat{\pi}_9 \widehat{\pi}_{\text{red}} \\
\widehat{\pi}(\varphi) &= \widehat{\pi}(\psi) + \widehat{\pi}(\chi) = \widehat{\pi}_1 \widehat{\pi}_{\text{blue}} + \widehat{\pi}_9 \widehat{\pi}_{\text{red}}
\end{aligned}
$$

*(2) Logical scores.*

$$
\begin{aligned}
s_t(c_i, \boldsymbol{x}) &= s_i(t, \boldsymbol{x}) - s_\emptyset(t, \boldsymbol{x}), \\
s_t(\psi, \boldsymbol{x}) &= s_t(1, \boldsymbol{x}) + s_t(\text{blue}, \boldsymbol{x}) \\
s_t(\chi, \boldsymbol{x}) &= s_t(9, \boldsymbol{x}) + s_t(\text{red}, \boldsymbol{x}) \\
s_t(\varphi, \boldsymbol{x}) &= \frac{\widehat{\pi}(\psi) s_t(\psi, \boldsymbol{x}) + \widehat{\pi}(\chi) s_t(\chi, \boldsymbol{x})}{\widehat{\pi}(\psi) + \widehat{\pi}(\chi)}
\end{aligned}
$$

Substituting the posterior coefficients and logical scores gives the fully expanded logical score:

$$\boxed{s_t(\varphi, \boldsymbol{x}) = \frac{\widehat{\pi}_1 \widehat{\pi}_{\text{blue}} \big(s_t(1, \boldsymbol{x}) + s_t(\text{blue}, \boldsymbol{x})\big) + \widehat{\pi}_9 \widehat{\pi}_{\text{red}} \big(s_t(9, \boldsymbol{x}) + s_t(\text{red}, \boldsymbol{x})\big)}{\widehat{\pi}_1 \widehat{\pi}_{\text{blue}} + \widehat{\pi}_9 \widehat{\pi}_{\text{red}}}}$$

All $\widehat{\pi}_i$ are scalar coefficients, and all spatial derivatives come from the diffusion model via $s_t(c_i, \boldsymbol{x}) = s_i - s_\emptyset$. The final score is calculated by adding the logical score to the unconditional score: $s_\emptyset(t, \boldsymbol{x}) + s_t(\varphi, \boldsymbol{x})$.

*Repulsive Guidance.* For *repulsive guidance*, the atomic condition $c$ is extended to $c \wedge \neg c_{\text{not}}$, where $c_{\text{not}}$ denotes the most probable class (excluding $c$ itself). This introduces a repulsive term that explicitly discourages confusion with the dominant alternative. The modified atomic score becomes

$$s_t^{\text{RG}}(c, \boldsymbol{x}) = \big(s_c(t, \boldsymbol{x}) - s_\emptyset(t, \boldsymbol{x})\big) - \frac{\widehat{\pi}(c_{\text{not}})}{1 - \widehat{\pi}(c_{\text{not}})} \big(s_{c_{\text{not}}}(t, \boldsymbol{x}) - s_\emptyset(t, \boldsymbol{x})\big).$$

For example, when targeting the digit 1, we set $c = 1$ and choose $c_{\text{not}}$ as the most likely alternative digit (e.g., 9 if $\widehat{\pi}_9$ is maximal). The resulting score pushes samples away from features of 9 while pushing towards features of 1. In the case of a negation, repulsive guidance induces a double-negation effect: samples are pushed away from the negated class and additionally steered towards the most probable class. This modified atomic score can be used within the same recursive construction as above.

# C. Additional Theoretical Results

## C.1. Taxonomy Queries

In many practical settings, the atomic predicates are not unrelated attributes, but form a *taxonomy*. Typical examples include biological classifications (e.g., phylum, genus, species), object hierarchies in vision (e.g., vehicle, car, sedan), or semantic type systems. In such cases, predicates are either mutually exclusive (siblings in the taxonomy) or strictly nested (a child predicate implies its parent). Users often wish to specify constraints at different levels of this hierarchy, for example, allowing any instance of a broad category (e.g., mammal) or restricting generation to a small number of specific subcategories. The following proposition shows that any such taxonomy query, specified simply as a set of allowed nodes in the hierarchy, admits an exact logical guidance rule under our framework.

**Proposition C.1.** *Fix $t \in (0, T]$ and $\boldsymbol{x} \in \mathcal{X}$. Let $\mathcal{T}$ be a finite taxonomy of propositions, that is, a set of predicates $\{c_u : u \in V\}$ indexed by nodes of a rooted tree $(V, \preceq)$, with root $r$ such that*

$$c_r \equiv \top, \qquad and \qquad u \preceq v \;\Rightarrow\; [\![c_u]\!] \subseteq [\![c_v]\!],$$

*and siblings are mutually exclusive: if $u$ and $v$ are distinct children of the same parent, then $[\![c_u]\!] \cap [\![c_v]\!] = \emptyset$.*

*A* taxonomy query *is specified by a set of allowed nodes $A \subseteq V$, interpreted as the event*

$$\varphi_A := \bigvee_{u \in A} c_u.$$

*There exists a semantically equivalent formula $\psi_A$ whose circuit uses only $\neg$-nodes and OR-ME $\vee$-nodes. Consequently, the recursive rules in Table 1 compute $p_t(\varphi_A \mid \boldsymbol{x})$ and $s_t(\varphi_A, \boldsymbol{x})$ exactly.*

*Proof.* For each node $u \in V$, define its *exclusive refinement* ("exactly $u$ and none of its descendants")

$$r_u := c_u \wedge \neg\Big( \bigvee_{v \in \mathrm{ch}(u)} c_v \Big),$$

where $\mathrm{ch}(u)$ are the children of $u$. Because siblings are mutually exclusive, the disjunction $\bigvee_{v \in \mathrm{ch}(u)} c_v$ is OR-ME.

Moreover, the family $\{r_u : u \in V\}$ is pairwise mutually exclusive, and its union is $\top$: every terminal sample belongs to exactly one "most specific" taxonomy node along its root-to-leaf path. Hence any taxonomy query $\varphi_A = \bigvee_{u \in A} c_u$ is semantically equivalent to the (mutually exclusive) disjunction of all exclusive refinements contained in it:

$$\varphi_A' := \bigvee_{u \in V:\; [\![r_u]\!] \subseteq [\![\varphi_A]\!]} r_u. \tag{11}$$

The right-hand side is an OR-ME disjunction of subformulas built using only negation and OR-ME disjunctions, and we have $[\![\varphi_A]\!] = [\![\varphi_A']\!]$. Therefore $\varphi_A'$ admits a circuit satisfying Prop. 3.1, and exactness of the recursive rules follows. $\square$

## C.2. Completeness Conditions for Circuit Compilability

We presented in Proposition 3.1 a set of exact recursive composition rules. We now show that these recursive rules are complete for two classes of distributions of general interest.

The first class concerns properties that may be partitioned into multiple groups, where properties within the same group are mutually exclusive and those between groups are conditionally independent. This situation arises, for example, when we have multiple categorical properties, each of which are treated as independent. For instance, we may have color and digit-class as the groups, and the properties $\{\mathrm{red}, \mathrm{green}, \mathrm{blue}\}$ in the former group and $\{0, ..., 9\}$ in the latter. We note that this general case also covers multiple conditionally independent binary properties, since the groups are singletons.

The second class concerns sets of properties that are strictly nested, hence for any two properties, we have either that they are mutually exclusive, or that one property implies the other. This case covers the situation where we have a strictly nested ontology, for instance, a taxonomy of microbial species, genera and phyla.

To demonstrate completeness, we introduce the notion of the *compilability* of a Boolean formula $\varphi$. We write $PC(\varphi)$ for the probabilistic circuit formed by directly parsing $\varphi$, so that conjunctions of subformulas are mapped to conjunction nodes,

and similarly for disjunctions and negations. $PC(\varphi)$ provides a direct representation of $\varphi$, and the function $PC(.)$ provides a one-to-one mapping between Boolean formulas and probabilistic circuits. However, the circuit $PC(\varphi)$ may be inadequate for certain purposes; in particular, it may not satisfy the conditions of Proposition 3.1. A *compilation* of $\varphi$, denoted $PC^*(\varphi)$, allows $PC(\varphi)$ to be further manipulated according to a compilation scheme, to produce a circuit $PC^*(\varphi) = PC(\varphi')$ with desirable properties, which maintains the semantics of the original Boolean formula. Hence, we must have $[\![\varphi]\!] = [\![\varphi']\!]$. A formula $\varphi$ is compilable for a given distribution and class of valid circuits iff $PC^*(\varphi)$ belongs to the class of valid circuits.

In the following proposition, we show that the logical composition rules we provide in Table 1 are complete for the classes of distribution noted above, in the sense that any Boolean formula $\varphi$ may be compiled into a probabilistic circuit fulfilling the conditions of Proposition 3.1, which can therefore be exactly evaluated. We note that, for the second class of distribution, all formulas compile to probabilistic circuits in a more restricted set (excluding the $\wedge$-CI rule), and we have the additional guarantee that, if these conditions are satisfied at terminal time ($t = 0$), they are satisfied for all $\boldsymbol{x}_t$ and $t > 0$.

**Proposition C.2.** *Given a distribution over $\boldsymbol{x}_0$ conditioned on $\boldsymbol{X}_t = \boldsymbol{x}_t$ with $t \in (0, T]$, and predicates $\{c_1, ...c_N\}$ over $\boldsymbol{x}_0$, for any Boolean formula $\varphi$ over the same predicates such that $[\![\varphi]\!] \neq [\![\bot]\!]$, we have that $\varphi$ may be compiled into a probabilistic circuit evaluable by the rules of Table 1 at $\boldsymbol{X}_t = \boldsymbol{x}_t$ when the atomic predicates satisfy:*

1. *The predicates fall into subsets, $S_1, ..., S_M$, where $S_m \subseteq \{1...N\}$, $m_1 \neq m_2 \Rightarrow S_{m_1} \cap S_{m_2} = \emptyset$ and $\cup_m S_m = \{1...N\}$, such that (a) $n_1, n_2 \in S_m$ and $n_1 \neq n_2$ implies that the events $c_{n_1}$ and $c_{n_2}$ are ME given $\boldsymbol{X}_t = \boldsymbol{x}_t$, (b) for all tuples $(n_{m=1}, ..., n_{m=M})$ such that $\forall m.(n_m \in S_m)$, we have that the events $c_{n_1}, ..., c_{n_M}$ are mutually CI given $\boldsymbol{X}_t = \boldsymbol{x}_t$, and (c) for all $m$, $p(\neg \vee_{n \in S_m} c_n) = 0$.*

*Moreover, we have that $\varphi$ may be compiled into a probabilistic circuit evaluable by the rules of Table 1 at all $t \geq 0$ and settings $\boldsymbol{X}_t = \boldsymbol{x}_t$ when, at terminal time ($t = 0$), the atomic predicates satisfy:*

2. *For each pair of predicates, $c_i$ and $c_j$ ($i \neq j$), we have that $[\![c_i]\!] \cap [\![c_j]\!] \in \{\emptyset, [\![c_i]\!], [\![c_j]\!]\}$ (i.e., they are ME or nested).*

*Proof.* We consider the two cases in the proposition separately.

*Case 1:* For case (1) of Proposition C.2, we consider an arbitrary predicate, $\varphi$. We may write $\varphi$ in Full Disjunctive Normal Form (FDNF) as:

$$\varphi \equiv \text{FDNF}(\varphi) := \bigvee_{i \in R(\varphi)} \left( \bigwedge_{j \in \{1...N\}} y_{ij} \right) \tag{12}$$

where $y_{ij} = \neg c_j$ if $\text{mod}(i, 2^j) < 2^{j-1}$ and $y_{ij} = c_j$ otherwise; and $R(\varphi) \subseteq \{0...(2^N - 1)\}$ such that $n \in R(\varphi) \iff \varphi \wedge (\bigwedge_j y_{nj}) \not\equiv \bot$. We note that Equation (12) is a disjunction of subformulas, $Y_i = \bigwedge_j y_{ij}$, each of which is, by definition, mutually exclusive. By the assumptions of case (1) of the proposition, if, for a given $i$, there exists an $m$ such that $(\sum_{j \in S_i} \mu(i, j)) \neq 1$, where $\mu(i, j) = [\text{mod}(i, 2^j) \geq 2^{j-1}]$ and $[.]$ denotes the Iverson bracket, we must have $[\![Y_i]\!] = [\![\bot]\!]$. Hence, we may consider $R'(\varphi) = R(\varphi) \backslash \{i | \exists m.(\sum_{n \in S_m} \mu(i, j)) \neq 1\}$. Moreover, since for any $i \in R'(\varphi)$, there is exactly one $j$ from any set $S_m$ such that $\mu(i, j) = 1$, we may consider the predicate:

$$\varphi' := \bigvee_{i \in R'(\varphi)} \left( \bigwedge_{m \in \{1...M\}} Y'_{im} \right) \tag{13}$$

where $Y'_{im} = c_{j(i,m)}$ such that $j(i, m)$ is the unique $j \in S_m$ such that $\mu(i, j) = 1$. We thus have $[\![\varphi]\!] = [\![\text{FDNF}(\varphi)]\!] = [\![\varphi']\!]$, where the first equality holds by definition, and the second holds since $\varphi'$ is formed by excluding only those terms from the disjunction $\text{FDNF}(\varphi)$ corresponding to the event $\emptyset$. We observe that $\varphi'$ is a categorical-CI query, as defined in Section 3.1, since conditions $(a)$ and $(c)$ in the proposition ensure that the properties $\{c_n | n \in S_m\}$ may be treated as values in the domain of a categorical variable, only one of which may be selected, and condition $(b)$ ensures that the categorical variables $1...M$ are mutually CI, since for any configuration $(c_{n_m} = 1, ..., c_{n_M} = 1)$ where $\forall m.(n_m \in S_m)$, we have $p(c_{n_m} = 1, ..., c_{n_M} = 1) = \prod_m p(c_{n_m} = 1)$. Hence $PC(\varphi')$ may be evaluated exactly according to our framework, following Proposition 3.2. $\varphi$ may thus be compiled to the circuit $PC^*(\varphi) = PC(\varphi')$, which can be evaluated exactly as required.

*Case 2:* For case (2) of Proposition C.2, we have a Boolean formula $\varphi$ over atoms $\mathcal{C} = \{c_1, ..., c_N\}$ satisfying the condition that any two distinct atoms are ME or nested. We first consider an expansion of the original system to $\mathcal{C}^\dagger \supseteq \mathcal{C}$. To form $\mathcal{C}^\dagger$, we begin by adding an atom $c_\top$ which is semantically equivalent to $\top$, if no such atom exists in $\mathcal{C}$; this preserves the property in (2), since all other atoms must be nested inside $\top$. Further, for every atom $c_{1...N}$, we add a new atom $c'_n$, corresponding to the event:

$$\llbracket c'_n \rrbracket = \llbracket c_n \wedge \neg(\vee_{m \in C_n} c_m) \rrbracket \tag{14}$$

where $C_n = \{m \leq N | \llbracket c_n \rrbracket \supset \llbracket c_m \rrbracket \wedge \neg \exists l \leq N. \llbracket c_n \rrbracket \supset \llbracket c_l \rrbracket \supset \llbracket c_m \rrbracket\}$, for all $n \leq N$ except those for which $\llbracket c'_n \rrbracket = \emptyset$. The expanded system $\mathcal{C}^\dagger$ will retain the property in (2), since for all $c'_n$, we have that $c'_n$ is nested inside $c_n$, and for any other atom $c_m$ in $\mathcal{C}^\dagger$, if $c_n$ is ME or nested inside $c_m$, $c'_n$ will likewise be ME or nested inside $c_m$ respectively, and if $c_m$ is nested inside $c_n$, $c_m$ and $c'_n$ will be ME. We observe that the model defined over $\mathcal{C}^\dagger$ is equivalent in expressive power to $\mathcal{C}$, since any formula $\varphi$ over $\mathcal{C}$ is also a formula over $\mathcal{C}^\dagger$, and any formula $\phi$ over $\mathcal{C}^\dagger$ may be mapped to a semantically equivalent formula $\varphi = K(\phi)$ over $\mathcal{C}$ by replacing all occurrences of $c_\top$ with $\top$, and all occurrences of $c'_n$ with $c_n \wedge \neg(\vee_{m \in C_n} c_m)$.

We now consider compiling an arbitrary Boolean circuit $\varphi$ over $\mathcal{C}$ given the assumption in (2). We first lift $\varphi$ to a formula $\phi$ over $\mathcal{C}^\dagger$; as noted, we may simply set $\phi = \varphi$. We now consider $\text{FDNF}(\phi)$, defined analogously to Eq. 12 (where we note that we now have $N^\dagger = |\mathcal{C}^\dagger|$ atoms). For each $i$ let $J'_i = \{j | \mu(i, j) = 1\}$. If $\exists j \in J_i$ s.t. $\forall j' \in J_i$, $\llbracket c^\dagger_j \wedge c^\dagger_{j'} \rrbracket = \llbracket c^\dagger_j \rrbracket$, and further that $\neg(\exists j' \notin J_i. \llbracket c^\dagger_j \wedge c^\dagger_{j'} \rrbracket = \llbracket c^\dagger_{j'} \rrbracket)$, then we have $\llbracket Y_i \rrbracket = \llbracket c^\dagger_{j(i)} \rrbracket$, where we write $j(i)$ for the unique such $j$. For all other $i$, we have $\llbracket Y_i \rrbracket = \llbracket \bot \rrbracket$, since either there must exist $j, j' \in J_i$, $j \neq j'$ such that $c^\dagger_j$ and $c^\dagger_{j'}$ are ME, or $\forall j' s.t. \llbracket c^\dagger_j \wedge c^\dagger_{j'} \rrbracket = \llbracket c^\dagger_{j'} \rrbracket . j' \notin J_i$. Hence, we may consider $R''(\phi) = R(\phi) \backslash \{i | \exists j \in J_i . (\forall j' \in J_i . \llbracket c^\dagger_j \wedge c^\dagger_{j'} \rrbracket = \llbracket c^\dagger_j \rrbracket) \wedge \neg(\exists j' \notin J_i . \llbracket c^\dagger_j \wedge c^\dagger_{j'} \rrbracket = \llbracket c^\dagger \rrbracket_{j'})\}$. We now consider the predicate:

$$\phi' := \bigvee_{i \in R''(\phi)} c^\dagger_{j(i)} \tag{15}$$

By definition, $\llbracket \phi \rrbracket = \llbracket \phi' \rrbracket = \llbracket \varphi \rrbracket$. We also have that $PC(\phi')$ is a *taxonomy query* as defined above, and so is exactly evaluable in our framework by Proposition C.1. However, $\phi'$ is defined in the expanded system $\mathcal{C}^\dagger$, and so it may include atoms not in the original system. We therefore map $\phi'$ to a predicate $\varphi' = K(\phi')$ over $\mathcal{C}$ using the transformation $K(.)$ defined above. This will result in a formula $\varphi'$ which is a disjunction of ME subformulas, where the latter are all either atoms or negations of disjunctions of ME atoms (since $C_n$ contains only the direct children of $c_n$, which by definition cannot be nested). $\varphi'$ is thus evaluable in our framework, since all nodes in $PC(\varphi')$ are either $\neg$ or $\vee$-MI nodes (we note that $\wedge$-CI and $\vee$-CI are not necessary). Hence, $\varphi$ may be compiled to the circuit $PC^*(\varphi) = PC(\varphi')$ (we note that, alternatively, we may form $PC^*(\varphi)$ by applying $K(.)$ to the formula constructed for the evaluation of $\phi'$ as a taxonomy query in the proof of Proposition C.1, (Equation (11)), which will likewise result in a circuit in which all nodes are either $\neg$ or $\vee$-MI). Finally, we have that, if $c_i$ and $c_j$ are ME ($\llbracket c_i \wedge c_j \rrbracket = \emptyset$) at terminal time $t = 0$, they will be ME at all $\boldsymbol{x}$ and $t > 0$. Hence, it is sufficient that (2) is satisfied at terminal time for $\varphi$ to be compilable at all $t$ and $\boldsymbol{X}_t = \boldsymbol{x}_t$, as required.

$\square$

## C.3. Discrete Logical Guidance

We briefly consider the here case in which, instead of score functions for the unconditional and atomic conditional models, we instead are supplied with transition functions for these generative processes for a set of discrete time-steps. The underlying space $\mathcal{X}$ may be either continuous or discrete. We denote the unconditional transition function as $\tau_t(\boldsymbol{x}_{t-1} | \boldsymbol{x}_t)$, and the transition function conditioned on predicate $c_i$ as $\tau_t(\boldsymbol{x}_{t-1} | c_i, \boldsymbol{x}_t)$. Then, under the same structural assumptions as Proposition 3.1, $\tau_t(\boldsymbol{x}_{t-1} | \varphi, \boldsymbol{x}_t)$ and $\widehat{\pi}(\varphi)$ may be calculated by mutual recursion, where the rules for calculating $\widehat{\pi}(\varphi)$ are as in Table 1, and the rules for calculating $\tau_t(\boldsymbol{x}_{t-1} | \varphi, \boldsymbol{x}_t)$ follow an analogous pattern to those for $s(\varphi, \boldsymbol{x})$ in Table 1:

$$\tau_t(\boldsymbol{x}_{t-1}|\boldsymbol{x}_t), \tau_t(\boldsymbol{x}_{t-1}|c_i, \boldsymbol{x}_t) \qquad \text{provided}$$

$$\tau_t(\boldsymbol{x}_{t-1}|\neg\psi, \boldsymbol{x}_t) = \frac{\tau_t(\boldsymbol{x}_{t-1}|\boldsymbol{x}_t) - \widehat{\pi}(\psi)\tau_t(\boldsymbol{x}_{t-1}|\psi, \boldsymbol{x}_t)}{1 - \widehat{\pi}(\psi)},$$

$$\tau_t^{\mathrm{CI}}(\boldsymbol{x}_{t-1}|\psi \wedge \chi, \boldsymbol{x}_t) = \frac{\tau_t(\boldsymbol{x}_{t-1}|\psi, \boldsymbol{x}_t)\tau_t(\boldsymbol{x}_{t-1}|\chi, \boldsymbol{x}_t)}{\tau_t(\boldsymbol{x}_{t-1}|\boldsymbol{x}_t)},$$

$$\tau_t^{\mathrm{ME}}(\boldsymbol{x}_{t-1}|\psi \vee \chi, \boldsymbol{x}_t) = \frac{\widehat{\pi}(\psi)\tau_t(\boldsymbol{x}_{t-1}|\psi, \boldsymbol{x}_t) + \widehat{\pi}(\chi)\tau_t(\boldsymbol{x}_{t-1}|\chi, \boldsymbol{x}_t)}{\widehat{\pi}(\psi) + \widehat{\pi}(\chi)},$$

$$\tau_t^{\mathrm{CI}}(\boldsymbol{x}_{t-1}|\psi \vee \chi, \boldsymbol{x}_t) = \frac{\widehat{\pi}(\psi)\tau_t(\boldsymbol{x}_{t-1}|\psi, \boldsymbol{x}_t) + \widehat{\pi}(\chi)\tau_t(\boldsymbol{x}_{t-1}|\chi, \boldsymbol{x}_t) - \widehat{\pi}(\psi)\widehat{\pi}(\chi)\tau_t^{\mathrm{CI}}(\boldsymbol{x}_{t-1}|\psi \wedge \chi, \boldsymbol{x}_t)}{\widehat{\pi}(\psi) + \widehat{\pi}(\chi) - \widehat{\pi}(\psi)\widehat{\pi}(\chi)}.$$

$$(16)$$

We thus have the following discrete equivalent of Proposition 3.1:

**Proposition C.3.** *Let $\varphi$ be a propositional formula over atoms $\{c_i\}$. Suppose that $\varphi$ admits a circuit representation whose internal nodes are $\wedge$, $\vee$, and $\neg$, whose $\wedge$- and $\vee$-nodes satisfy conditions (1) and (2) from Proposition 3.1, and that that $\widehat{\pi}(\psi) = p_t(\psi \mid \boldsymbol{x})$ for all subformulas $\psi$ of $\varphi$. Then the recursive rules above in Equation (16) reproduce exactly the true posterior and conditional transition functions for $\varphi$:*

$$\widehat{\pi}(\varphi) = p_t(\varphi \mid \boldsymbol{x}), \qquad \tau_t(\boldsymbol{x}_{t-1}|\varphi, \boldsymbol{x}_t) = p(\boldsymbol{x}_{t-1}|\varphi, \boldsymbol{x}_t).$$

*Proof.* We argue by structural induction on the circuit of $\varphi$. Since the inductive steps for $\widehat{\pi}(.)$ are identical to those in Proposition 3.1, we omit them from the proof below.

*Base case (atoms).* For an atomic predicate $c_i$ we have by assumption that $\tau_t(\boldsymbol{x}_{t-1}|c_i, \boldsymbol{x}_t)$ are provided. Thus the claim holds.

*Inductive step (negation).* Suppose the claim holds for $\psi$. Then by definition we have:

$$\begin{aligned} p(\boldsymbol{x}_{t-1} \mid \neg\psi) &= \frac{p(\neg\psi \mid \boldsymbol{x}_{t-1}, \boldsymbol{x}_t)\tau_t(\boldsymbol{x}_{t-1}|\boldsymbol{x}_t)}{1 - \widehat{\pi}(\psi)} \\ &= \frac{(1 - p(\psi \mid \boldsymbol{x}_{t-1}, \boldsymbol{x}_t)))\tau_t(\boldsymbol{x}_{t-1}|\boldsymbol{x}_t)}{1 - \widehat{\pi}(\psi)} \\ &= \frac{\tau_t(\boldsymbol{x}_{t-1}|\boldsymbol{x}_t) - \tau_t(\boldsymbol{x}_{t-1}|\boldsymbol{x}_t)p(\psi \mid \boldsymbol{x}_{t-1}, \boldsymbol{x}_t))}{1 - \widehat{\pi}(\psi)} \\ &= \frac{\tau_t(\boldsymbol{x}_{t-1}|\boldsymbol{x}_t) - \widehat{\pi}(\psi)\tau_t(\boldsymbol{x}_{t-1}|\psi, \boldsymbol{x}_t)}{1 - \widehat{\pi}(\psi)} \end{aligned}$$

which matches the recursive rule for $\tau_t(\boldsymbol{x}_{t-1}|\neg\psi, \boldsymbol{x}_t)$.

*Inductive step (conjunction).* Let $\varphi = \psi \wedge \chi$ and assume the inductive hypothesis for $\psi$ and $\chi$. By assumption, $p_t(\psi \wedge \chi \mid \boldsymbol{x}) = p_t(\psi \mid \boldsymbol{x})p_t(\chi \mid \boldsymbol{x})$; hence, we have:

$$\begin{aligned} p(\boldsymbol{x}_{t-1} \mid \psi \wedge \chi) &= \frac{p(\psi \wedge \chi \mid \boldsymbol{x}_{t-1}, \boldsymbol{x}_t)\tau_t(\boldsymbol{x}_{t-1}|\boldsymbol{x}_t)}{p(\psi \wedge \chi \mid \boldsymbol{x}_t)} \\ &= \frac{p(\psi \mid \boldsymbol{x}_{t-1}, \boldsymbol{x}_t)p(\chi \mid \boldsymbol{x}_{t-1}, \boldsymbol{x}_t)\tau_t(\boldsymbol{x}_{t-1}|\boldsymbol{x}_t)}{\widehat{\pi}(\psi)\widehat{\pi}(\chi)} \\ &= \frac{\tau_t(\boldsymbol{x}_{t-1}|\psi, \boldsymbol{x}_t)\tau_t(\boldsymbol{x}_{t-1}|\boldsymbol{x}_t)}{\tau_t(\boldsymbol{x}_{t-1}|\boldsymbol{x}_t)} \end{aligned}$$

which is exactly the recursive rule for $\tau_t(\boldsymbol{x}_{t-1}|\psi \wedge \chi, \boldsymbol{x}_t)$.

*Inductive step (disjunction).* Let $\varphi = \psi \vee \chi$ and assume the inductive hypothesis for $\psi$ and $\chi$. By inclusion–exclusion,

$$p_t(\psi \vee \chi \mid \boldsymbol{x}) = p_t(\psi \mid \boldsymbol{x}) + p_t(\chi \mid \boldsymbol{x}) - p_t(\psi \wedge \chi \mid \boldsymbol{x}).$$

There are two cases.

If $\psi$ and $\chi$ are mutually exclusive, we have:

$$
\begin{aligned}
p(\boldsymbol{x}_{t-1} \mid \psi \vee \chi) &= \frac{p(\psi \vee \chi \mid \boldsymbol{x}_{t-1}, \boldsymbol{x}_t)\tau_t(\boldsymbol{x}_{t-1}|\boldsymbol{x}_t)}{p(\psi \vee \chi \mid \boldsymbol{x}_t)} \\
&= \frac{\left(p(\psi \mid \boldsymbol{x}_{t-1}, \boldsymbol{x}_t) + p(\chi \mid \boldsymbol{x}_{t-1}, \boldsymbol{x}_t)\right)\tau_t(\boldsymbol{x}_{t-1}|\boldsymbol{x}_t)}{\widehat{\pi}(\psi) + \widehat{\pi}(\chi)} \\
&= \frac{\widehat{\pi}(\psi)\tau_t(\boldsymbol{x}_{t-1}|\psi, \boldsymbol{x}_t) + \widehat{\pi}(\chi)\tau_t(\boldsymbol{x}_{t-1}|\chi, \boldsymbol{x}_t)}{\widehat{\pi}(\psi) + \widehat{\pi}(\chi)}
\end{aligned}
$$

which is exactly the recursive rule for $\tau_t^{\mathrm{ME}}(\boldsymbol{x}_{t-1}|\psi \vee \chi, \boldsymbol{x}_t)$.

If $\psi$ and $\chi$ are conditionally independent, we have:

$$
\begin{aligned}
p(\boldsymbol{x}_{t-1} \mid \psi \vee \chi) &= \frac{p(\psi \vee \chi \mid \boldsymbol{x}_{t-1}, \boldsymbol{x}_t)\tau_t(\boldsymbol{x}_{t-1}|\boldsymbol{x}_t)}{p(\psi \vee \chi \mid \boldsymbol{x}_t)} \\
&= \frac{\left(p(\psi \mid \boldsymbol{x}_{t-1}, \boldsymbol{x}_t) + p(\chi \mid \boldsymbol{x}_{t-1}, \boldsymbol{x}_t) - p(\psi \mid \boldsymbol{x}_{t-1}, \boldsymbol{x}_t)p(\chi \mid \boldsymbol{x}_{t-1}, \boldsymbol{x}_t)\right)\tau_t(\boldsymbol{x}_{t-1}|\boldsymbol{x}_t)}{\widehat{\pi}(\psi) + \widehat{\pi}(\chi) - \widehat{\pi}(\psi)\widehat{\pi}(\chi)} \\
&= \frac{\widehat{\pi}(\psi)\tau_t(\boldsymbol{x}_{t-1}|\psi, \boldsymbol{x}_t) + \widehat{\pi}(\chi)\tau_t(\boldsymbol{x}_{t-1}|\chi, \boldsymbol{x}_t) - \widehat{\pi}(\psi)\widehat{\pi}(\chi)\tau_t^{\mathrm{CI}}(\boldsymbol{x}_{t-1}|\psi \wedge \chi, \boldsymbol{x}_t)}{\widehat{\pi}(\psi) + \widehat{\pi}(\chi) - \widehat{\pi}(\psi)\widehat{\pi}(\chi)}
\end{aligned}
$$

which is exactly the recursive rule for $\tau_t^{\mathrm{CI}}(\boldsymbol{x}_{t-1}|\psi \vee \chi, \boldsymbol{x}_t)$.

In all cases, the recursive rules reproduce $\tau_t(\boldsymbol{x}_{t-1}|\varphi, \boldsymbol{x}_t)$ at the node $\varphi$ assuming they do so for its children. By structural induction over the circuit of $\varphi$, the claim holds for all subformulas and in particular for $\varphi$ itself. $\qquad \square$

Since the conditions on the circuit representation of $\varphi$ are the same in Proposition C.3 as Proposition 3.1, an analogous completeness result can be shown for the discrete case to Proposition C.2. This is shown by substituting the rules in Equation (16) for those of 1, and the conditions in Proposition C.3 for those of Proposition 3.1 in the statement and proof of C.2. This shows that the rules in Equation (16) achieve complete compilability for all Boolean formulas under the same conditions (Proposition C.2) in the discrete case as the continuous case.

# D. Implementation Details

## D.1. Sampling Implementation

We build our method upon standard diffusion backbones. For simpler benchmarks (e.g., CMNIST), we utilize the Denoising Diffusion Probabilistic Models (DDPM) (Ho et al., 2020). For large-scale experiments on ImageNet, we adopt the EDM2 framework by Karras et al. (Karras et al., 2024).

The sampling procedure, summarized in Algorithm 1, follows the standard reverse diffusion process with a modification analogous to CFG. The logical guidance score is calculated in Algorithm 2 recursively implementing the derived rules as stated in Table 1. Note that while Algorithm 1 and 1 use $\epsilon$-prediction instead of scores, the composition rules remain unchanged, as $\epsilon$ and the score differ only by a time-dependent scalar factor (Karras et al., 2022).

---

**Algorithm 1** Sampling Implementation with Logical Composition

1: **Input:** Query $\psi$, Diffusion model $\epsilon_\theta$, Classifier $p_\phi$, Timesteps $T$

2: $\mathbf{x}_T \sim \mathcal{N}(\mathbf{0}, \mathbf{I})$
3: **for** $t = T, \dots, 1$ **do**
4:     $\hat{\epsilon}_\emptyset \leftarrow \epsilon_\theta(\mathbf{x}_t, \emptyset, t)$     *// Unconditional score*

5:     *// Guide with Logical Score*
6:     $\mathbf{g}_{t,\psi, \_} \leftarrow \text{GETLOGICALSCORE}(\psi, \mathbf{x}_t)$     *// Algorithm 2*
7:     $\tilde{\epsilon}_t \leftarrow \hat{\epsilon}_\emptyset + \mathbf{g}_{t,\psi}$     *// Add logical posterior score to unconditional score*

8:     $\mathbf{x}_{t-1} \leftarrow \text{SAMPLERSTEP}(\mathbf{x}_t, \tilde{\epsilon}_t, t)$
9: **end for**
10: **return** $\mathbf{x}_0$

---

**Practical Implementation Details.** While Algorithm 1 and 2 outlines the general logic, our practical implementation introduces specific adjustments for numerical stability. All probability computations are performed in log-space to avoid underflow. Additionally, the probability-dependent task-specific guidance scales are clamped to a maximum value of 3. Finally, the global guidance weight is applied directly at the condition level, scaling each conditioning term individually before aggregation. The output of the unconditional model is cached to avoid repeated evaluation.

**Conditioning.** During training, we use single-feature condition: each sample is conditioned on one attribute while all other attributes are masked (e.g., "1, None" for a blue 1 in CMNIST). At inference, individual conditions are accessed via null-tokenization of the other attributes, allowing the conditional scores $s_c(t, \boldsymbol{x})$ to be computed per atomic condition. While methods such as CoInd (Gaudi et al., 2025) can enforce conditional independence when training on multiple attributes, we adopt single-feature training to isolate the effects of our compositional rules and ensure conditional independence in this work.

## D.2. Diffusion Architecture, Training, Sampling

Table 9 details the model architectures and hyperparameters for the CMNIST, Shapes3D, and CelebA experiments. For ImageNet-512, we utilize the pre-trained EDM2-XXL model (Karras et al., 2024), adopting the parameters corresponding to the best reported FID ($g = 2.05$, $\text{EMA}_m = 0.075$, $\text{EMA}_g = 0.155$). We note that our reproduced FID result (with no repulsive guiding) deviate slightly from the original publication, which we attribute to the unavailability of the exact sampling seeds.

## D.3. Classifier Architecture and Training

Table 10 details the architectures and hyperparameters for CMNIST, Shapes3D, and CelebA classifiers. We use two variants: *Judge* classifiers, trained on clean images to compute CS, and *Composition* classifiers, trained on noisy images to estimate condition probabilities $\hat{\pi}$ for compositional guidance.

---

**Algorithm 2** Recursive Logical Score Calculation

---

1: **Input:** Query $\psi$, Noisy state $\mathbf{x}_t$, Classifier $p_\phi$, Score model $\epsilon_\theta$, Guidance scale $w_{\text{guidance}}$, Repulsive guidance scale $w_{\text{not}}$

2: *// Returns tuple of logical score and probability $(\mathbf{g}_t, \hat{\pi})$ as stated in Table 1*
3: **function** GETLOGICALSCORE($\psi$, $\mathbf{x}_t$)

4:     *// 1. Atomic Condition (Base Case)*
5:     **if** $\varphi$ is **Atom** $c$ **then**
6:         $\pi_c \leftarrow p_\psi(c \mid \mathbf{x}_t)$
7:         $\mathbf{g}_{t,c} \leftarrow w_{\text{guidance}} \cdot \big(\epsilon_\theta(\mathbf{x}_t, c) - \hat{\epsilon}_\theta(\mathbf{x}_t, \emptyset)\big)$

9:         **if** doRepulsiveGuiding **then**
10:             $c_{\text{not}} \leftarrow \arg\max_{k \in \mathcal{C} \setminus \{c\}} \pi_k$     *// Find most probable condition $c_{not}$ s.t. $c_{not} \neq c$*
11:             $\mathbf{g}_{t,c} \leftarrow w_{\text{guidance}} \cdot \big(\epsilon_\theta(\mathbf{x}_t, c) - \hat{\epsilon}_\theta(\mathbf{x}_t, \emptyset)\big) - w_{\text{not}} \cdot \frac{\pi_{c_{\text{not}}}}{1 - \pi_{c_{\text{not}}}} \cdot \big(\epsilon_\theta(\mathbf{x}_t, c_{\text{not}}) - \hat{\epsilon}_\theta(\mathbf{x}_t, \emptyset)\big)$
12:         **end if**
13:         **return** $(\mathbf{g}_{t,c}, \pi_c)$
14:     **end if**

15:     *// 2. Recursive Composition*
16:     *// Negation ($\neg$)*
17:     **if** $\varphi$ is $\neg\psi$ **then**
18:         $(\mathbf{g}_{t,\psi}, \pi_\psi) \leftarrow$ GETLOGICALSCORE($\psi$, $\mathbf{x}_t$)
19:         $\pi_\varphi \leftarrow 1 - \pi_\psi$
20:         $\mathbf{g}_{t,\varphi} \leftarrow -\frac{\pi_\psi}{1 - \pi_\psi} \cdot \mathbf{g}_{t,\psi}$
21:         **return** $(\mathbf{g}_{t,\varphi}, \pi_\varphi)$
22:     **end if**

23:     *// Conjunction ($\wedge$)*
24:     **if** $\varphi$ is $\psi \wedge \chi$ **then**
25:         $(\mathbf{g}_{t,\psi}, \pi_\psi) \leftarrow$ GETLOGICALSCORE($\psi$, $\mathbf{x}_t$)
26:         $(\mathbf{g}_{t,\chi}, \pi_\chi) \leftarrow$ GETLOGICALSCORE($\chi$, $\mathbf{x}_t$)
27:         $\pi_\varphi \leftarrow \pi_\psi \cdot \pi_\chi$
28:         $\mathbf{g}_{t,\varphi} \leftarrow \mathbf{g}_{t,\psi} + \mathbf{g}_{t,\chi}$
29:         **return** $(\mathbf{g}_{t,\varphi}, \pi_\varphi)$
30:     **end if**

31:     *// Disjunction ($\vee$)*
32:     **if** $\varphi$ is $\psi \vee \chi$ **then**
33:         $(\mathbf{g}_{t,\psi}, \pi_\psi) \leftarrow$ GETLOGICALSCORE($\psi$, $\mathbf{x}_t$)
34:         $(\mathbf{g}_{t,\chi}, \pi_\chi) \leftarrow$ GETLOGICALSCORE($\chi$, $\mathbf{x}_t$)
35:         **if** Mutually Exclusive ($\vee_{\text{ME}}$) **then**
36:             $\pi_\varphi \leftarrow \pi_\psi + \pi_\chi$
37:             $w_\psi \leftarrow \frac{\pi_\psi}{\pi_\varphi + \pi_\chi}, \quad w_\chi \leftarrow \frac{\pi_\chi}{\pi_\varphi + \pi_\chi}$
38:         **else if** Conditional Independent ($\vee_{\text{CI}}$) **then**
39:             $\pi_\varphi \leftarrow \pi_\psi + \pi_\chi - \pi_\psi \pi_\chi$
40:             $w_\psi \leftarrow \frac{\pi_\psi(1 - \pi_\chi)}{\pi_\varphi + \pi_\chi - \pi_\varphi \pi_\chi}, \quad w_\chi \leftarrow \frac{\pi_\chi(1 - \pi_\psi)}{\pi_\varphi + \pi_\chi - \pi_\varphi \pi_\chi}$
41:         **end if**
42:         $\mathbf{g}_{t,\varphi} \leftarrow w_\psi \cdot \mathbf{g}_{t,\psi} + w_\chi \cdot \mathbf{g}_{t,\chi}$
43:         **return** $(\mathbf{g}_{t,\varphi}, \pi_\varphi)$
44:     **end if**
45: **end function**

---

*Table 9.* Hyperparameters for CMNIST, Shapes3D, and CelebA models.

| Hyperparameter | CMNIST | Shapes3D | CelebA |
|---|---|---|---|
| **Optimization** | | | |
| Optimizer | AdamW | AdamW | AdamW |
| Learning Rate | $1.0 \times 10^{-3}$ | $2.0 \times 10^{-4}$ | $2.0 \times 10^{-4}$ |
| Weight Decay | $1.0 \times 10^{-5}$ | 0.0 | – |
| LR Scheduler | Cosine w/ Warmup | Cosine w/ Warmup | Cosine w/ Warmup |
| Warmup Steps | 5000 | 5000 | 2000 |
| Num Training Steps | 50,000 | 100,000 | 500,000 |
| **Diffusion** | | | |
| Noise Scheduler | DDPM | DDPM | DDPM |
| Beta Schedule | Linear | Linear | Squared Cos Cap v2 |
| Train Timesteps | 1000 | 1000 | 1000 |
| Prediction Type | $\epsilon$ (epsilon) | $\epsilon$ (epsilon) | $\epsilon$ (epsilon) |
| **Model Architecture** | | | |
| Model Type | U-Net | U-Net | SiT (Transformer) |
| Input Size | $28 \times 28$ | $64 \times 64$ | $16 \times 16$ (Latent) |
| Channels / Hidden Dim | [56, 112, 168] | [56, 112, 168, 224] | 384 |
| Layers | 2 per block | 2 per block | 12 (Depth) |
| Attention | Head Dim: 8 | Head Dim: 8 | Heads: 6 |
| Dropout | 0.1 | 0.1 | – |
| Norm Groups | 8 | 8 | – |
| Activation | GELU | SiLU | – |
| Patch Size | – | – | 1 |
| **Sampling & Inference** | | | |
| Sampler | DDPM | DDPM | DDIM |
| Timesteps | 50 | 50 | 1000 |
| Guidance scale | 1.0 | 1.0 | 1.0 |

## D.4. Classifier from diffusion model

To approximate the probabilities $p(c_i|x)$ we can estimate (Li et al., 2023) the probabilities or their ratio using the pre-trained diffusion model $\epsilon_t(x, c_i)$, in particular, we can estimate the probability of the class $c_i$ given the current sample $x_t$ as

$$p(c_i|x_t) = \frac{1}{1 + \sum_{j \neq i} \exp \mathbb{E}_{\epsilon \sim \mathcal{N}(0,1)} [\|\epsilon - \epsilon_t(x_{t-1}, c_i)\|^2 - \|\epsilon - \epsilon_t(x_{t-1}, c_j)\|^2]}$$

where the noisy samples are generated either from the current samples $x_t$ and the noise $\epsilon \sim \mathcal{N}(0, 1)$,

$$\boldsymbol{x}_{t-1} = \sqrt{\alpha_t}\boldsymbol{x}_t + \sqrt{1 - \alpha_t}\epsilon$$

We can also estimate the ratio of the probability and its complement

$$\gamma(c_i|x_t) = \frac{p(c_i|x_t)}{1 - p(c_i|x_t)} = \frac{1}{\sum_{j \neq i} \exp \mathbb{E}_{\epsilon \sim \mathcal{N}(0,1)} [\|\epsilon - \epsilon_t(x_{t-1}, c_i)\|^2 - \|\epsilon - \epsilon_t(x_{t-1}, c_j)\|^2]}$$

## D.5. Unconditional Model from score functions

If the pre-trained diffusion model is not trained as unconditional model, then we can estimate the score of the uncondtional model from the conditional score. Let's suppose that we only have two classes $A, B$, we then marginalize and derive the unconditional distribution

$$p(\boldsymbol{x}) = p(\boldsymbol{x}|A)p(A) + p(\boldsymbol{x}|B)p(B)$$

*Table 10.* Hyperparameters for CMNIST, Shapes3D, and CelebA Classifiers.

| Hyperparameter | CMNIST | Shapes3D | CelebA |
|---|---|---|---|
| **Optimization** | | | |
| Optimizer | AdamW | AdamW | AdamW |
| Learning Rate | $3.0 \times 10^{-4}$ | $3.0 \times 10^{-4}$ | $3.0 \times 10^{-4}$ |
| Weight Decay | $1.0 \times 10^{-3}$ | $1.0 \times 10^{-3}$ | $1.0 \times 10^{-3}$ |
| LR Scheduler | Cosine Annealing | Cosine Annealing | Cosine Annealing |
| Epochs (Judge) | 50 | 50 | 10 |
| Epochs (Composition) | 50 | 50 | 100 |
| **Model Architecture** | | | |
| Model type | ResNet18 | ResNet18 | ResNet18 |
| Classes per Label | [10, 10] | [10, 10, 10, 8, 4, 15] | [2, 5] |
| **Noise Injection (Composition only)** | | | |
| Noise Scheduler | DDPM | DDPM | DDPM |
| Train Timesteps | 1000 | 1000 | 1000 |
| Beta Schedule | Linear | Linear | Squared Cos Cap v2 |

We can then estimate the unconditional probability score function as

$$\nabla \ln p(\boldsymbol{x}) = \frac{\nabla p(\boldsymbol{x}|A)p(A) + \nabla p(\boldsymbol{x}|B)p(B)}{p(\boldsymbol{x}|A)p(A) + p(\boldsymbol{x}|B)p(B)}$$

$$= \frac{p(\boldsymbol{x}|A)p(A)}{p(\boldsymbol{x}|A)p(A) + p(\boldsymbol{x}|B)p(B)}\nabla \ln p(\boldsymbol{x}|A) + \frac{p(\boldsymbol{x}|B)p(B)}{p(\boldsymbol{x}|A)p(A) + p(\boldsymbol{x}|B)p(B)}\nabla \ln p(\boldsymbol{x}|B)$$

$$= \frac{p(A|\boldsymbol{x})p(\boldsymbol{x})}{p(A|\boldsymbol{x})p(\boldsymbol{x}) + p(B|\boldsymbol{x})p(\boldsymbol{x})}\nabla \ln p(\boldsymbol{x}|A) + \frac{p(\boldsymbol{x}|B)p(B)}{p(\boldsymbol{x}|A)p(A) + p(\boldsymbol{x}|B)p(B)}\nabla \ln p(\boldsymbol{x}|B).$$

After simplification we have that

$$\nabla \ln p(\boldsymbol{x}) = \frac{p(A|\boldsymbol{x})}{p(A|\boldsymbol{x}) + p(B|\boldsymbol{x})}\nabla \ln p(\boldsymbol{x}|A) + \frac{p(B|\boldsymbol{x})}{p(A|\boldsymbol{x}) + p(B|\boldsymbol{x})}\nabla \ln p(\boldsymbol{x}|B),$$

If we then defined $\{c_i\} = \{A, B\}$, we can write the score function in a compact form as

$$\nabla \ln p(\boldsymbol{x}) = \text{soft-max}_{p(c_i|\boldsymbol{x})}\nabla \ln p(\boldsymbol{x}|c_i)$$

If we have a finite number of classes $\{c_i\}$, we can write the unconditional probability as

$$p(\boldsymbol{x}) = \sum_i p(\boldsymbol{x}|c_i)p(c_i),$$

and following the previous derivation, we have that

$$\boxed{\nabla \ln p(\boldsymbol{x}) = \text{soft-max}_{p(c_i|\boldsymbol{x})}\nabla \ln p(\boldsymbol{x}|c_i)} \tag{17}$$

# E. Experiment details

### E.1. Task Generation

Queries are generated via a recursive algorithm where the complexity parameter, $N$ expressions, defines the total number of binary operators. At each step, an operator is chosen uniformly at random from $\land$, $\lor_{CI}$, $\lor_{ME}$, determining how attributes are distributed to child nodes:

- AND / OR-CI: The available attribute groups are partitioned into two disjoint sets. The left and right terms are restricted to mutually exclusive sets of attributes (e.,g., the left queries colors, while the right queries digits).

- OR-ME: Both left and right expressions are constrained to share the same attribute group (e. g., both query Color).

Logical negation is applied probabilistically (p=0.05) to any node. For CMNIST an example of a query with $N = 4$ is: $\left( \left( \neg 6 \lor_{ME} (5 \lor_{ME} 3) \right) \lor_{CI} (\text{red} \lor_{ME} \text{yellow}) \right)$.

### E.2. Metrics

**Conformity score.** We quantify generation accuracy using the Conformity Score (Gaudi et al., 2025), defined as the percentage of samples that comply with the logical query. Sample attributes are inferred using a ResNet-18 (He et al., 2016) classifier (per-attribute accuracies reported in Table 11) and matched against the query logic. We report the average conformity score calculated over 10,000 samples, generated from 100 unique queries per task type with 100 samples each.

*Table 11.* Feature-wise accuracy of the classifiers used to calculate Conformity Score and Joint Entropy.

| Dataset | Feature | Accuracy (%) |
|---|---|---|
| CMNIST | Digit | 98.41 |
| | Color | 100 |
| Shapes3D | Floor Hue | 100 |
| | Wall Hue | 100 |
| | Object Hue | 100 |
| | Scale | 100 |
| | Shape | 100 |
| | Orientation | 100 |
| CelebA | Gender | 98.00 |
| | Hair color | 89.28 |

**Joint entropy (Diversity).** A key requirement for controlled generation is that the model covers the full scope of valid solutions without mode collapse. For example, given the CMNIST query $(9 \lor 3)$, the model should generate nines and threes across all available colors, rather than collapsing to a narrow subset.

To quantify this, we calculate the Mean Batch Joint Entropy. Evaluation is performed per batch ($n = 100$ images) where the query is fixed. For every image $i$, we extract the attribute vector using the oracle classifier. We treat these vectors as outcomes of a joint random variable and calculate the Shannon entropy based on the counts of unique attribute combinations within the batch.

Let $C$ be the set of unique attribute combinations observed in the batch, and $p(c)$ be the frequency of a specific combination $c \in C$. The joint entropy is defined as:

$$H(Batch) = -\sum_{c \in C} p(c) \log_2 p(c).$$

Since the number of valid solutions varies by query (e. g., restrictive queries naturally permit lower diversity), the theoretical optimal entropy is task-dependent. We therefore include the theoretical optimal entropy in Figure 3 as a reference baseline to contextualize the model's performance.

**Fréchet Inception Distance.** To assess image fidelity and distributional alignment, we calculate the FID. For CelebA, we compute the FID independently for each compositional task. We generate 5,000 samples per task (comprising 100 images for each of 50 queries) and utilize the clean-fid implementation (Seitzer, 2020). For ImageNet, we adopt the EDM2 framework (Karras et al., 2024), calculating FID over 50,000 generated samples.

# F. Ligand-Protein Multi-target structure-based drug design

Following Zhou et al. (2024); Skreta et al. (2025a), we first align in 3D the target protein pockets of the two targets. We then apply the logic guidance of the $SO(3)$-equivariant graph neural network to generate the ligand over 1000 time steps or denoisy steps. For each experiments we generated 32 samples of size 23. At the end of the generation, we use AutoDock Vina to compute the main metrics. We use $\beta = 2.0$, the inverse temperature during the generation. We also generate the single target baseline using the TargetDiff (Guan et al., 2023). In the experiments we used GRM5-RRM1 (with UniProt IDs: P41594 and P23921) protein pairs. To evaluate the performance of the generated ligand we considered QED and SA scores. The quality indicator (Lee et al., 2025) checks that the drug-likeness (QED) $\geq 0.6$ and the synthetic accessibility (SA) $\leq 4.0$. Experimental code for the Multi-Target drug design is provided at https://github.com/nec-research/Logical-Guidance-for-the-Exact-Composition-of-Diffusion-Models.

## F.1. Interaction profile of representative designed ligands

To illustrate the molecular basis of selectivity induced by logical compositions, we analyzed the 2D interaction profiles of representative ligands generated by LOGDIFF (Figure 6). The ligands were chosen as follows: (i) For $A \wedge B$, we selected a ligand where Vina's binding score was the lowest for both targets; (ii) For $A \wedge \neg B$, we selected a ligand with the lowest score for target A and the highest score for target B. The reference ligand establishes hydrogen bonds with key residues in both binding sites: Ser590A and Ser488A in GRM5, and Val805A and Tyr315A in RRM1. In the representative example of a ligand generated under the $A \wedge B$ constraint, this dual interaction pattern is preserved, with the ligand forming contacts with His372A, Leu592A, and Pro133A in GRM5, while simultaneously engaging Thr804A, Glu319A, and Val805A in RRM1. In contrast, the representative ligand generated with the $A \wedge \neg B$ constraint exhibits a markedly distinct interaction profile: it maintains multiple hydrophobic and hydrogen bond contacts with GRM5 (Phe602A, Val378A, Thr499A, Gly377A, Ser590A) while showing no detectable interactions with RRM1. These representative examples illustrate that the LOGDIFF framework not only satisfies logical constraints in terms of docking scores but can also generate ligands with chemically distinct interaction profiles, supporting the applicability of logical composition for selective drug design.

We note that AutoDock Vina scores serve here as a proxy for binding affinity, as they incorporate physical terms (van der Waals interactions, hydrogen bonding, hydrophobic contacts, and torsional entropy) that provide a physically grounded, if approximate, characterization of ligand–protein interactions. While sufficient to demonstrate the qualitative effect of logical constraints on binding selectivity, we acknowledge that more rigorous assessments, such as free energy perturbation (FEP), thermodynamic integration (TI), or ultimately experimental validation, would be necessary to draw conclusions about true binding affinity. The present results should therefore be interpreted as a proof-of-concept demonstration of the logical guidance framework in the molecular domain.

## F.2. Additional target proteins

We report the results with additional targets;s pairs in this session. We consider 14 target pairs as used in previous work. The pairs are $(8, 226)$, $(29, 371)$, $(200, 416)$, $(164, 287)$, $(208, 209)$, $(31, 313)$, $(373, 398)$, $(226, 8)$, $(371, 29)$, $(416, 200)$, $(287, 164)$, $(209, 208)$, $(313, 31)$, $(398, 373)$, where the number is the index of the protein from the dataset. The pairs are symmetric, from the results, we can observe that the generation protocol is not symmetric. The results on the 14 targets' pairs are reported in Table 12, and Table 13. While Table 14 and Table 15 show the effect of the FKC correction. We use 4 runs per pair and generate ligand of length 23.

## F.3. Per Pair analysis of ligand generation

We report the performance of the composition of generative models in generating candidates ligands for 14 targets' pairs separately. In Table 16 and Table 17, we show how the generation without the FKC correction, which is reported in Table 18, and Table 19. We notice that the baseline method with the negative logic generate invalid candidates for some pairs of target proteins. We also observe that the performance with symmetric target pairs is relative different. We also notice that the proposed approach does not always produce the best performance, even in the previous analysis we observe that the mean performance is improved.

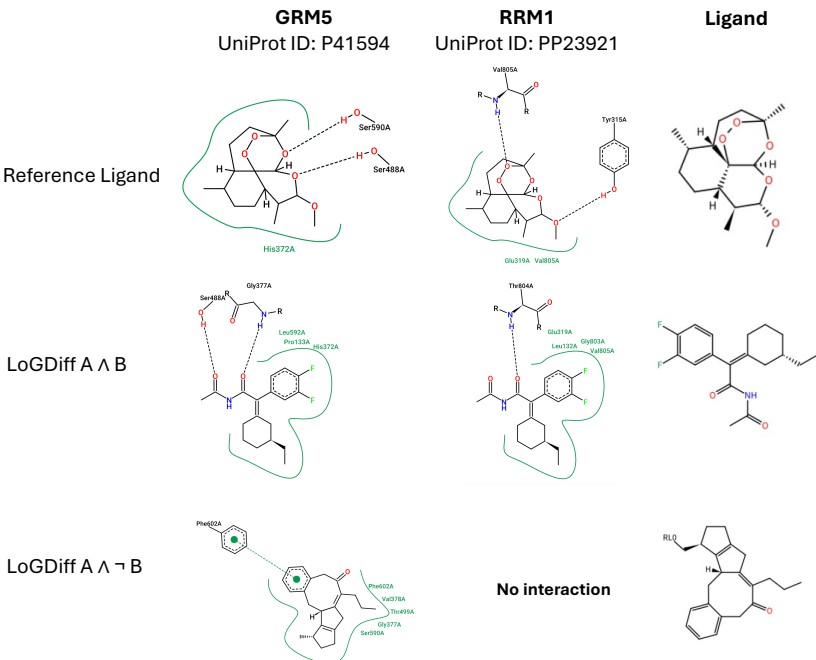

*Figure 6.* **2D interaction analysis of representative ligands generated for the GRM5-RRM1 target pair.** (Top) Reference ligand showing hydrogen bonds with key residues in both targets. (Middle) Representative ligand generated under the conjunction constraint $(A \land B)$, displaying hydrogen bonds and hydrophobic contacts with residues from both GRM5 and RRM1. (Bottom) Representative ligand generated under the selective constraint $(A \land \neg B)$, maintaining multiple interactions with GRM5 while showing no detectable interactions with RRM1. Green contours indicate hydrophobic regions; black dashed lines represent hydrogen bonds, whereas green dashed lines represenr $\pi$-$\pi$ interactions. Images were rendered with ProteinPlus.

## G. Baselines and additional results

### G.1. Baseline implementations

We compare our approach against prior methods for logical composition in diffusion models, in particular Liu et al. (2022) and Skreta et al. (2025b). The exact composition rules used for each method are summarized in Table 20.

**Relation to compositional sculpting (Garipov et al., 2023).** The method of Garipov et al. (2023) formulates composition in diffusion models via binary operators, in particular the harmonic mean and a contrast operator, implemented through classifier-guided sampling. While effective for pairwise combinations, this framework is inherently limited to binary compositions. Notably, the contrast operator does not correspond to a logical disjunction, but is instead more closely related to a conjunction with negation (i.e., $A \land \neg B$). Even in the binary case, realizing a logical OR would require prior knowledge of the relative probabilities of the component distributions, which are generally not available in practice. Moreover, the formulation systematically suppresses overlapping regions, which is inconsistent with the semantics of a logical OR that preserves such regions. In addition, the method requires training an auxiliary classifier to distinguish between the component distributions involved in each composition (e.g., between $p_1$ and $p_2$ in a binary operator). As a result, extending the approach to settings with many attributes or more complex compositions would require a large number of such classifiers, limiting its scalability. For these reasons, we do not consider it as a suitable baseline for our setting.

### G.2. Compute time comparison

Table 21 reports the average batch sampling time for different composition operations and methods. For LoGDIFF, runtime scales approximately linearly with the number of diffusion model evaluations required by the logical query. For example, a conjunction with LoGDIFF requires three evaluations, corresponding to the unconditional score and the two conditional scores. This explains the increase in runtime for recursive queries with larger $N$. Moreover, LoGDIFF and Liu et al. (2022) require nearly the same compute time for the operations supported by both methods, despite LoGDIFF using classifier-based weights instead of constant weights. This indicates that sampling time is dominated by diffusion model evaluations rather

*Table 12.* Evaluation of LOGDIFF to generate drug candidates with dual on-targets for additional 14 targets, along with the baselines: TargetDiff (Guan et al., 2023), and DualDiff (Yang et al., 2024). A higher average docking score should correlate with a higher binding affinity to both targets.

| | (A * B) ($\uparrow$) | A ($\downarrow$) | B ($\downarrow$) | Div. ($\uparrow$) | Val. & Uniq. ($\uparrow$) | Qual. ($\uparrow$) |
|---|---|---|---|---|---|---|
| TargetDiff | $65.82_{\pm14.37}$ | $-8.76_{\pm0.97}$ | $-7.44_{\pm1.43}$ | $0.89_{\pm0.01}$ | $0.95_{\pm0.05}$ | $0.16_{\pm0.10}$ |
| | | | $A \wedge B$ | | | |
| DualDiff | $76.21_{\pm12.21}$ | $-8.74_{\pm0.92}$ | $-8.66_{\pm0.82}$ | $0.89_{\pm0.01}$ | $0.97_{\pm0.06}$ | $0.21_{\pm0.12}$ |
| LOGDIFF | $77.29_{\pm11.52}$ | $-8.79_{\pm0.89}$ | $-8.75_{\pm0.80}$ | $0.89_{\pm0.01}$ | $0.98_{\pm0.04}$ | $0.15_{\pm0.10}$ |
| | | | $A \vee B$ | | | |
| LOGDIFF | $77.27_{\pm11.91}$ | $-8.77_{\pm0.89}$ | $-8.76_{\pm0.81}$ | $0.89_{\pm0.01}$ | $0.98_{\pm0.04}$ | $0.15_{\pm0.10}$ |

*Table 13.* Evaluation of LOGDIFF with on and off targets on 14 target pairs, along with the baselines: TargetDiff and DualDiff. The $A \wedge \neg B$ uses $A$ as on-target and $B$ as off-target; in the $A \oplus B$, the roles of on- and off-targets are left to the generative model.

| | $\Delta$(A, B) ($\uparrow$) | A | B | Div. ($\uparrow$) | Val. & Uniq. ($\uparrow$) | Qual. ($\uparrow$) |
|---|---|---|---|---|---|---|
| | | | $A \wedge \neg B$ | | | |
| DualDiff | $1.57_{\pm1.08}$ | $-13.92_{\pm2.17}$ | $-14.01_{\pm1.76}$ | $0.87_{\pm0.02}$ | $0.64_{\pm0.24}$ | $0.00_{\pm0.00}$ |
| SDE | $0.99_{\pm0.48}$ | $-8.79_{\pm0.84}$ | $-8.63_{\pm0.73}$ | $0.89_{\pm0.01}$ | $0.97_{\pm0.08}$ | $0.17_{\pm0.13}$ |
| LOGDIFF | $0.99_{\pm0.51}$ | $-8.80_{\pm0.85}$ | $-8.65_{\pm0.81}$ | $0.89_{\pm0.01}$ | $0.96_{\pm0.09}$ | $0.14_{\pm0.10}$ |
| | | | $A \oplus B$ | | | |
| LOGDIFF | $1.07_{\pm0.59}$ | $-8.79_{\pm0.89}$ | $-8.74_{\pm0.80}$ | $0.89_{\pm0.01}$ | $0.98_{\pm0.04}$ | $0.15_{\pm0.10}$ |

than by the lightweight classifiers used to estimate the composition weights. Compared to Skreta et al. (2025b), LOGDIFF is substantially more efficient, while additionally providing composition rules for recursive logical queries that are not defined by the baselines.

### G.3. Comparison with additional versions and effect of probability estimator

We compare our approach against existing frameworks that define logical composition for diffusion models through linear combinations of scores (Liu et al., 2022; Skreta et al., 2025b). For Liu et al. (2022), we evaluate conjunction both with and without the unconditional model. To assess the effect of the estimator proposed by Skreta et al. (2025b) on disjunctions, we also evaluate their formula using a classifier. We report the comparative results in Table 22. As a lower bound, we include an unconditional baseline, which represents the probability of satisfying the logical constraints by chance.

For conjunctions, we observe that simply averaging conditional scores or re-weighting individual scores to achieve equal density (Skreta et al., 2025b) performs considerably worse than our method and the posterior-score-based approach of Liu et al. (2022). For disjunctions, our method and the re-weighting scheme proposed by Skreta et al. (2025b) achieve comparable results, independently of whether the estimator or the classifier is used.

For negation, the constant baseline achieves slightly higher conformity scores than LOGDIFF, though this comes at a the cost to diversity and perceptual quality (see Section 4.1). The EBM-style negation $p(x)/p_A(x)^\gamma$ (Du et al., 2023), requires exhaustive optimization of the hyperparameter $\gamma$. Even after tuning the guidance weight to maximize CS while maintaining sample validity, performance remains only marginally better than the unconditional baseline.

### G.4. Qualitative effect of guidance weight

Figure 7 qualitatively illustrates the trade-off between conformity and diversity as the guidance weight $w$ increases which is quantitatively reported in Figure 3. While higher guidance scales improve conformity scores, they visibly reduce sample diversity (see Figure 7).

*Table 14.* Evaluation of LOGDIFF to generate drug candidates with dual on-targets on 14 targets' pairs, along with the baselines: TargetDiff (Guan et al., 2023), and DualDiff (Yang et al., 2024). A higher average docking score should correlate with a higher binding affinity to both targets.

| | $(A * B)$ ($\uparrow$) | A ($\downarrow$) | B ($\downarrow$) | Div. ($\uparrow$) | Val. & Uniq. ($\uparrow$) | Qual. ($\uparrow$) |
|---|---|---|---|---|---|---|
| | | | $A \wedge B$ **(FKC)** | | | |
| LOGDIFF | $71.96_{\pm 12.99}$ | $-8.51_{\pm 0.89}$ | $-8.40_{\pm 0.87}$ | $0.78_{\pm 0.08}$ | $0.89_{\pm 0.17}$ | $0.26_{\pm 0.21}$ |
| | | | $A \vee B$ **(FKC)** | | | |
| LOGDIFF | $70.22_{\pm 13.03}$ | $-8.30_{\pm 0.98}$ | $-8.40_{\pm 0.90}$ | $0.78_{\pm 0.06}$ | $0.93_{\pm 0.15}$ | $0.21_{\pm 0.22}$ |

*Table 15.* Evaluation of LOGDIFF with on and off targets on 14 targets' pairs, along with the baselines: TargetDiff and DualDiff. The $A \wedge \neg B$ uses $A$ as on-target and $B$ as off-target; in the $A \oplus B$, the roles of on- and off-targets are left to the generative model.

| | $\Delta(A, B)$ ($\uparrow$) | A | B | Div. ($\uparrow$) | Val. & Uniq. ($\uparrow$) | Qual. ($\uparrow$) |
|---|---|---|---|---|---|---|
| | | | $A \wedge \neg B$ **(FKC)** | | | |
| SDE | $0.72_{\pm 0.33}$ | $-8.38_{\pm 0.94}$ | $-8.30_{\pm 0.83}$ | $0.79_{\pm 0.06}$ | $0.86_{\pm 0.23}$ | $0.21_{\pm 0.21}$ |
| LOGDIFF | $0.81_{\pm 0.41}$ | $-8.44_{\pm 0.86}$ | $-8.31_{\pm 0.83}$ | $0.78_{\pm 0.06}$ | $0.90_{\pm 0.17}$ | $0.24_{\pm 0.21}$ |
| | | | $A \oplus B$ **(FKC)** | | | |
| LOGDIFF | $0.85_{\pm 0.54}$ | $-8.53_{\pm 0.97}$ | $-8.42_{\pm 0.90}$ | $0.78_{\pm 0.07}$ | $0.91_{\pm 0.10}$ | $0.23_{\pm 0.19}$ |

## G.5. Qualitative results of logical compositions on images

We present qualitative results across various datasets and logical composition tasks, comparing our approach against the constant baseline. Each figure displays a batch of samples for a given query, generated with a guidance weight of $w = 1$. Consistent with the quantitative results in Table 2, our method performs identically to the baseline for conjunctions. While the constant baseline achieves higher conformity scores on negation tasks, this comes at the cost of reduced diversity. Furthermore, the baseline struggles with disjunctions and complex logical statements.

*Table 16.* Evaluation of LoGDiff to generate drug candidates with dual on-targets for additional 14 targets, along with the baselines: TargetDiff (Guan et al., 2023), and DualDiff (Yang et al., 2024). A higher average docking score should correlate with a higher binding affinity to both targets.

| (A * B) ($\uparrow$) | DualDiff $A \wedge B$ | TargetDiff | LoGDiff $A \wedge B$ | LoGDiff $A \vee B$ |
|---|---|---|---|---|
| (8,226) | $74.44_{\pm 1.14}$ | $70.65_{\pm 3.37}$ | $74.42_{\pm 0.34}$ | $74.22_{\pm 0.20}$ |
| (29,371) | $53.07_{\pm 2.41}$ | $56.12_{\pm 0.78}$ | $60.05_{\pm 1.67}$ | $59.95_{\pm 1.61}$ |
| (200,416) | $80.69_{\pm 5.70}$ | $80.89_{\pm 0.98}$ | $80.15_{\pm 9.50}$ | $77.88_{\pm 11.16}$ |
| (164,287) | $69.72_{\pm 4.07}$ | $63.38_{\pm 0.83}$ | $69.51_{\pm 4.09}$ | $69.30_{\pm 4.21}$ |
| (208,209) | $86.00_{\pm 1.32}$ | $92.06_{\pm 3.08}$ | $84.00_{\pm 4.47}$ | $85.16_{\pm 3.51}$ |
| (31,313) | $89.76_{\pm 6.19}$ | $67.22_{\pm 3.18}$ | $94.47_{\pm 5.62}$ | $95.83_{\pm 5.21}$ |
| (373,398) | $73.89_{\pm 3.20}$ | $73.20_{\pm 1.16}$ | $80.77_{\pm 2.95}$ | $80.64_{\pm 3.01}$ |
| (226,8) | $68.96_{\pm 3.99}$ | $49.80_{\pm 23.91}$ | $73.39_{\pm 1.46}$ | $73.40_{\pm 1.30}$ |
| (371,29) | $54.20_{\pm 3.77}$ | $52.00_{\pm 6.76}$ | $58.71_{\pm 2.56}$ | $58.65_{\pm 2.43}$ |
| (416,200) | $82.28_{\pm 5.17}$ | $44.71_{\pm 8.80}$ | $79.14_{\pm 4.42}$ | $79.53_{\pm 4.04}$ |
| (287,164) | $72.29_{\pm 2.42}$ | $66.00_{\pm 2.42}$ | $69.17_{\pm 3.43}$ | $68.60_{\pm 3.67}$ |
| (209,208) | $85.21_{\pm 3.96}$ | $70.79_{\pm 1.35}$ | $88.16_{\pm 2.31}$ | $87.96_{\pm 1.90}$ |
| (313,31) | $89.14_{\pm 10.60}$ | $68.54_{\pm 1.50}$ | $89.00_{\pm 10.57}$ | $89.05_{\pm 10.77}$ |
| (398,373) | $76.67_{\pm 2.43}$ | $62.50_{\pm 2.53}$ | $78.38_{\pm 4.53}$ | $78.49_{\pm 4.49}$ |

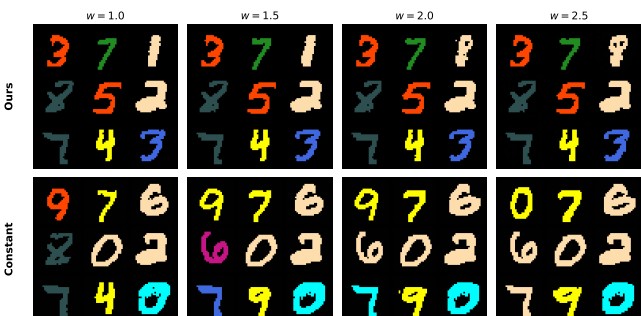

*Figure 7.* Influence of guidance weight $w$ on negation (query: $\neg$ 1). Higher guidance scales for the negation reduce the diversity of the constant baseline, while increasing the conformity score of our approach without harming diversity.

*Table 17.* Evaluation of LoGDiff with on and off targets on 14 target pairs, along with the baselines: TargetDiff and DualDiff. The $A \wedge \neg B$ uses $A$ as on-target and $B$ as off-target; in the $A \oplus B$, the roles of on- and off-targets are left to the generative model. $-$ indicates that the design did not convergence when generating the final molecule, this is also reflected by the Quality score.

| $\Delta$(A, B) ($\uparrow$) | DualDiff $A \wedge \neg B$ | SDE $A \wedge \neg B$ | LoGDiff $A \wedge \neg B$ | LoGDiff $A \oplus B$ |
|---|---|---|---|---|
| (8,226) | $-$ | $0.84_{\pm 0.05}$ | $0.91_{\pm 0.08}$ | $0.86_{\pm 0.19}$ |
| (29,371) | $-$ | $0.51_{\pm 0.03}$ | $0.43_{\pm 0.12}$ | $0.53_{\pm 0.06}$ |
| (200,416) | $2.42_{\pm 0.06}$ | $0.57_{\pm 0.13}$ | $0.59_{\pm 0.11}$ | $0.50_{\pm 0.05}$ |
| (164,287) | $0.22_{\pm 0.10}$ | $1.21_{\pm 0.18}$ | $1.21_{\pm 0.06}$ | $1.05_{\pm 0.07}$ |
| (208,209) | $1.61_{\pm 0.06}$ | $1.23_{\pm 0.23}$ | $1.30_{\pm 0.13}$ | $2.12_{\pm 0.42}$ |
| (31,313) | $3.65_{\pm 0.00}$ | $1.18_{\pm 0.41}$ | $0.83_{\pm 0.14}$ | $1.05_{\pm 0.34}$ |
| (373,398) | $1.85_{\pm 0.00}$ | $1.93_{\pm 0.36}$ | $2.10_{\pm 0.44}$ | $1.65_{\pm 0.36}$ |
| (226,8) | $0.15_{\pm 0.04}$ | $0.97_{\pm 0.14}$ | $0.92_{\pm 0.22}$ | $0.92_{\pm 0.11}$ |
| (371,29) | $-$ | $0.43_{\pm 0.08}$ | $0.38_{\pm 0.07}$ | $0.42_{\pm 0.09}$ |
| (416,200) | $2.13_{\pm 0.00}$ | $0.52_{\pm 0.10}$ | $0.56_{\pm 0.12}$ | $0.55_{\pm 0.09}$ |
| (287,164) | $0.50_{\pm 0.17}$ | $1.19_{\pm 0.24}$ | $1.32_{\pm 0.23}$ | $1.46_{\pm 0.48}$ |
| (209,208) | $1.54_{\pm 0.00}$ | $1.56_{\pm 0.58}$ | $1.36_{\pm 0.61}$ | $1.77_{\pm 0.56}$ |
| (313,31) | $3.47_{\pm 0.00}$ | $1.34_{\pm 0.42}$ | $1.09_{\pm 0.42}$ | $0.89_{\pm 0.09}$ |
| (398,373) | $1.50_{\pm 0.05}$ | $0.86_{\pm 0.09}$ | $0.98_{\pm 0.12}$ | $1.24_{\pm 0.27}$ |

*Table 18.* Evaluation of LOGDIFF to generate drug candidates with dual on-targets on 14 targets' pairs, along with the baselines: TargetDiff (Guan et al., 2023), and DualDiff (Yang et al., 2024). A higher average docking score should correlate with a higher binding affinity to both targets.

| (A * B) (↑) | TargetDiff | LOGDIFF $A \wedge B$ (FKC) | LOGDIFF $A \vee B$ (FKC) |
|---|---|---|---|
| (8,226) | $70.65_{\pm3.37}$ | $73.86_{\pm4.26}$ | $75.74_{\pm3.12}$ |
| (29,371) | $56.12_{\pm0.78}$ | $63.80_{\pm4.89}$ | $64.84_{\pm3.19}$ |
| (200,416) | $80.89_{\pm0.98}$ | $79.35_{\pm25.02}$ | $69.69_{\pm10.36}$ |
| (164,287) | $63.38_{\pm0.83}$ | $65.38_{\pm4.62}$ | $65.65_{\pm4.04}$ |
| (208,209) | $92.06_{\pm3.08}$ | $94.28_{\pm4.87}$ | $90.38_{\pm6.50}$ |
| (31,313) | $67.22_{\pm3.18}$ | $72.49_{\pm5.62}$ | $77.29_{\pm6.56}$ |
| (373,398) | $73.20_{\pm1.16}$ | $72.15_{\pm4.31}$ | $61.72_{\pm9.23}$ |
| (226,8) | $49.80_{\pm23.91}$ | $69.58_{\pm13.11}$ | $64.08_{\pm9.41}$ |
| (371,29) | $52.00_{\pm6.76}$ | $55.49_{\pm1.90}$ | $53.47_{\pm4.98}$ |
| (416,200) | $44.71_{\pm8.80}$ | $87.80_{\pm2.16}$ | $76.54_{\pm5.11}$ |
| (287,164) | $66.00_{\pm2.42}$ | $69.14_{\pm2.97}$ | $67.17_{\pm3.67}$ |
| (209,208) | $70.79_{\pm1.35}$ | $85.13_{\pm9.76}$ | $88.98_{\pm7.60}$ |
| (313,31) | $68.54_{\pm1.50}$ | $63.51_{\pm1.33}$ | $57.61_{\pm8.19}$ |
| (398,373) | $62.50_{\pm2.53}$ | $62.85_{\pm6.37}$ | $67.24_{\pm12.85}$ |

*Table 19.* Evaluation of LOGDIFF with on and off targets on 14 targets' pairs, along with the baselines: TargetDiff and DualDiff. The $A \wedge \neg B$ uses $A$ as on-target and $B$ as off-target; in the $A \oplus B$, the roles of on- and off-targets are left to the generative model.

| $\Delta$(A, B) (↑) | SDE $A \wedge \neg B$ (FKC) | LOGDIFF $A \wedge \neg B$ (FKC) | LOGDIFF $A \oplus B$ (FKC) |
|---|---|---|---|
| (8,226) | $0.85_{\pm0.38}$ | $1.14_{\pm0.48}$ | $1.01_{\pm0.22}$ |
| (29,371) | $0.42_{\pm0.03}$ | $0.35_{\pm0.06}$ | $0.45_{\pm0.07}$ |
| (200,416) | $0.49_{\pm0.15}$ | $0.55_{\pm0.16}$ | $0.61_{\pm0.05}$ |
| (164,287) | $1.31_{\pm0.41}$ | $1.36_{\pm0.32}$ | $1.06_{\pm0.11}$ |
| (208,209) | $0.82_{\pm0.32}$ | $0.64_{\pm0.42}$ | $1.21_{\pm0.89}$ |
| (31,313) | $0.58_{\pm0.16}$ | $0.92_{\pm0.34}$ | $0.76_{\pm0.28}$ |
| (373,398) | $0.74_{\pm0.22}$ | $0.61_{\pm0.17}$ | $0.63_{\pm0.35}$ |
| (226,8) | $1.04_{\pm0.29}$ | $0.89_{\pm0.12}$ | $0.82_{\pm0.38}$ |
| (371,29) | $0.39_{\pm0.04}$ | $0.38_{\pm0.10}$ | $0.38_{\pm0.09}$ |
| (416,200) | $0.37_{\pm0.09}$ | $0.60_{\pm0.02}$ | $0.66_{\pm0.23}$ |
| (287,164) | $0.89_{\pm0.17}$ | $1.10_{\pm0.37}$ | $1.22_{\pm0.60}$ |
| (209,208) | $0.48_{\pm0.11}$ | $0.79_{\pm0.37}$ | $1.35_{\pm0.64}$ |
| (313,31) | $0.76_{\pm0.28}$ | $0.67_{\pm0.22}$ | $0.80_{\pm0.33}$ |
| (398,373) | $0.60_{\pm0.04}$ | $0.86_{\pm0.43}$ | $0.49_{\pm0.17}$ |

*Table 20.* **Logical composition rules across different Baselines**.

| Task | Method | Composition | use $\epsilon_\emptyset$ | Probabilities |
|---|---|---|---|---|
| AND | LOGDIFF | $\epsilon_\emptyset + (\epsilon_A - \epsilon_\emptyset) + (\epsilon_B - \epsilon_\emptyset)$ | Yes | - |
| | Liu et al. (2022) | $\epsilon_\emptyset + (\epsilon_A - \epsilon_\emptyset) + (\epsilon_B - \epsilon_\emptyset)$ | Yes | - |
| | Skreta et al. (2025b) | equal prob. update | No | Itô estimator |
| NOT | LOGDIFF | $\epsilon_\emptyset - \frac{p_A}{1-p_A}(\epsilon_A - \epsilon_\emptyset)$ | Yes | classifier |
| | Liu et al. (2022) | $\epsilon_\emptyset - (\epsilon_A - \epsilon_\emptyset)$ | Yes | - |
| OR-ME | LOGDIFF | $\epsilon_\emptyset + \frac{p_A}{p_A+p_B}(\epsilon_A - \epsilon_\emptyset) + \frac{p_B}{p_A+p_B}(\epsilon_B - \epsilon_\emptyset)$ | Yes | classifier |
| | Skreta et al. (2025b) | $\frac{p_A}{p_A+p_B}\epsilon_A + \frac{p_B}{p_A+p_B}\epsilon_B$ | No | Itô estimator |
| OR-CI | LOGDIFF | $\epsilon_\emptyset + \frac{p_A(1-p_B)(\epsilon_A-\epsilon_\emptyset)+p_B(1-p_A)(\epsilon_B-\epsilon_\emptyset)}{p_A+p_B-p_Ap_B}$ | Yes | classifier |
| | Skreta et al. (2025b) | $\frac{p_A}{p_A+p_B}\epsilon_A + \frac{p_B}{p_A+p_B}\epsilon_B$ | No | Itô estimator |

*Table 21.* Average sampling time per batch in seconds. Runtime primarily scales with the number of diffusion model evaluations required by each composition rule.

| Method | AND | NOT | OR-ME | OR-CI | $N = 2$ | $N = 3$ | $N = 4$ | $N = 5$ |
|---|---|---|---|---|---|---|---|---|
| **Shapes3D** | | | | | | | | |
| Uncond. | 7 | 7 | 7 | 7 | 7 | 7 | 7 | 7 |
| Skreta et al. (2025b) | 280 | - | 280 | 280 | - | - | - | - |
| Liu et al. (2022) | 20.8 | 13.9 | - | - | - | - | - | - |
| LOGDIFF | 20.8 | 14.0 | 21.1 | 21.1 | 28.2 | 35.3 | 42.6 | 49.9 |
| **CelebA** | | | | | | | | |
| Uncond. | 51.4 | 51.4 | 51.3 | 51.4 | | | | |
| Skreta et al. (2025b) | 1023.4 | - | 1023.2 | 1023.4 | - | - | - | - |
| Liu et al. (2022) | 151.8 | 101.6 | - | - | - | - | - | - |
| LOGDIFF | 151.8 | 102.6 | 155.6 | 155.9 | - | - | - | - |

*Table 22.* **Logical composition conformity across different frameworks on CMNIST**. We compare our method against logical different variant of composition frameworks and different probability estimators for diffusion models using linear combinations of scores (Liu et al., 2022; Du et al., 2023; Skreta et al., 2025b) using Conformity Scores (CS).

| Task | use $\epsilon_\emptyset$ | Probabilities | Method | CS |
|---|---|---|---|---|
| AND | Yes | - | $\epsilon_\emptyset$ only | 0.91 |
| | Yes | classifier | $\epsilon_\emptyset + (\epsilon_A - \epsilon_\emptyset) + (\epsilon_B - \epsilon_\emptyset)$ (LOGDIFF, Liu et al. (2022)) | **80.39** |
| | No | - | $\frac{1}{2}\epsilon_A + \frac{1}{2}\epsilon_B$ (Liu et al., 2022) | 26.61 |
| | No | Itô estimator | equal prob. update (Skreta et al., 2025b) | 26.20 |
| NOT | | | $\epsilon_\emptyset$ only | 90.19 |
| | Yes | - | $\epsilon_\emptyset - (\epsilon_A - \epsilon_\emptyset)$ (Liu et al., 2022) | **99.41** |
| | Yes | classifier | $\epsilon_\emptyset - \frac{p_A}{1-p_A}(\epsilon_A - \epsilon_\emptyset)$ (LOGDIFF) | 97.38 |
| | Yes | classifier | $\epsilon_\emptyset - w \cdot \epsilon_A$ (with $w = 0.07$) (Du et al., 2023) | 92.60 |
| OR-ME | | | $\epsilon_\emptyset$ only | 19.22 |
| | Yes | classifier | $\epsilon_\emptyset + \frac{p_A}{p_A+p_B}(\epsilon_A - \epsilon_\emptyset) + \frac{p_B}{p_A+p_B}(\epsilon_B - \epsilon_\emptyset)$ (LOGDIFF) | **98.01** |
| | No | classifier | $\frac{p_A}{p_A+p_B}\epsilon_A + \frac{p_B}{p_A+p_B}\epsilon_B$ (Skreta et al., 2025b) | 97.98 |
| | No | Itô estimator | $\frac{p_A}{p_A+p_B}\epsilon_A + \frac{p_B}{p_A+p_B}\epsilon_B$ (Skreta et al., 2025b) | 96.84 |
| OR-CI | | | $\epsilon_\emptyset$ only | 19.09 |
| | Yes | classifier | $\epsilon_\emptyset + \frac{p_A(1-p_B)(\epsilon_A-\epsilon_\emptyset)+p_B(1-p_A)(\epsilon_B-\epsilon_\emptyset)}{p_A+p_B-p_Ap_B}$ (LOGDIFF) | 97.25 |
| | No | classifier | $\frac{p_A}{p_A+p_B}\epsilon_A + \frac{p_B}{p_A+p_B}\epsilon_B$ (Skreta et al., 2025b) | 97.34 |
| | No | Itô estimator | $\frac{p_A}{p_A+p_B}\epsilon_A + \frac{p_B}{p_A+p_B}\epsilon_B$ (Skreta et al., 2025b) | **97.54** |

**light blue ∧ 3**

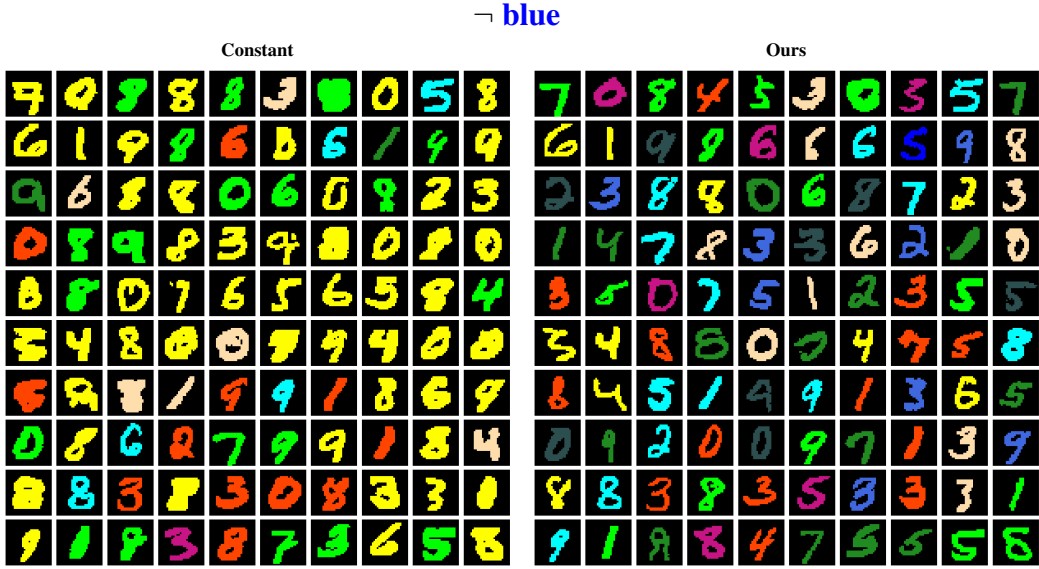

*Figure 8.* Batch of results for the AND composition (light blue ∧ 3) on CMNIST. Note that for this single AND operation, the constant baseline and our approach coincide. Conformity scores: Liu et al. (2022) = 0.79, LoGDIFF = 0.79.

**¬ blue**

*Figure 9.* Batch of results for the NOT composition (¬ blue) on CMNIST. Conformity scores: Liu et al. (2022) = 1.0, LoGDIFF = 0.99.

**light green floor ∧ blue wall**

Constant                                      Ours

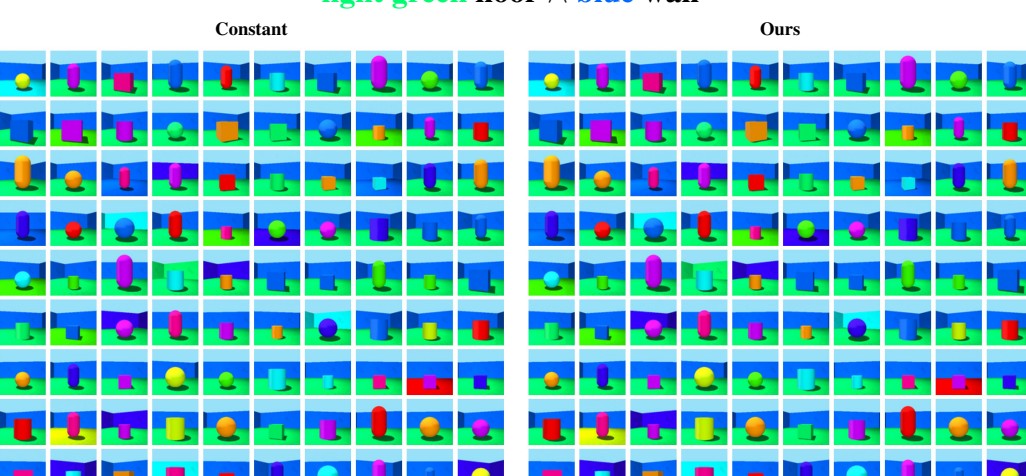

*Figure 10.* Batch of results for the AND composition (lightgreen floor ∧ blue wall) on Shapes3D. Conformity scores: Liu et al. (2022) = 0.74, LOGDIFF= 0.74.

**¬ cylinder**

Constant                                      Ours

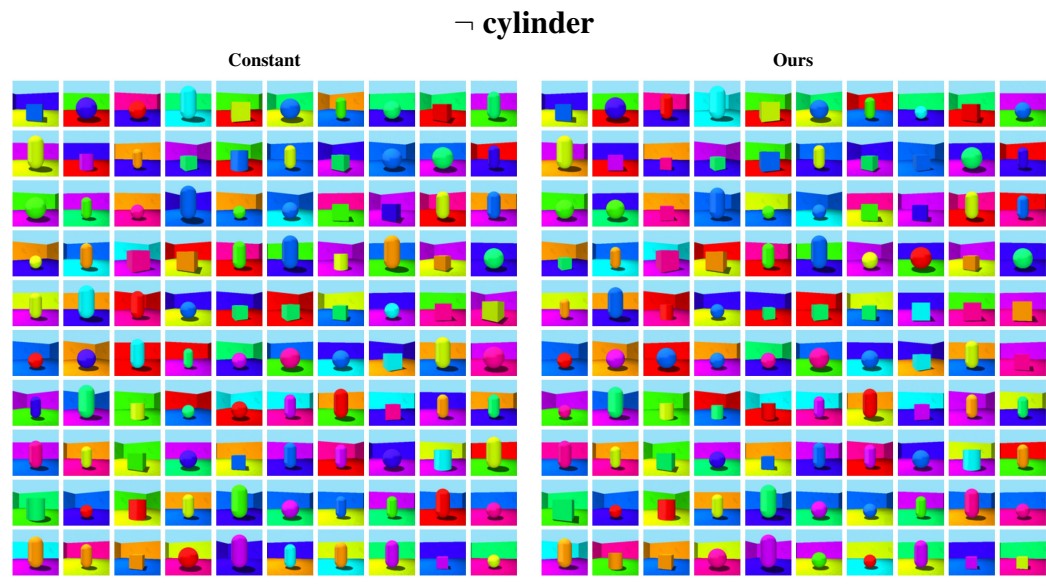

*Figure 11.* Batch of results for the NOT composition (¬ cylinder) on Shapes3D. Conformity scores: Liu et al. (2022) = 0.93, LOGDIFF= 0.93.

**((cube ∨$_{\text{CI}}$ (red wall ∨$_{\text{ME}}$ pink wall)) ∨$_{\text{CI}}$ (5 scale ∧ darkblue object))**

Ours

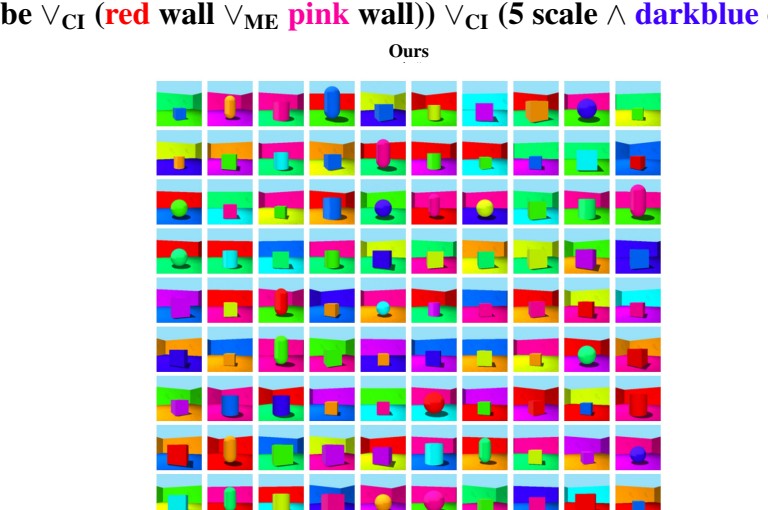

*Figure 12.* Batch of results for a complex composition with $N = 4$ ((cube ∨$_{\text{CI}}$ (red wall ∨$_{\text{ME}}$ pink wall)) ∨$_{\text{CI}}$ (5 scale ∧ darkblue object)) on Shapes3D. Conformity scores: LOGDIFF = 0.98.

**¬ blond**

Constant        Ours

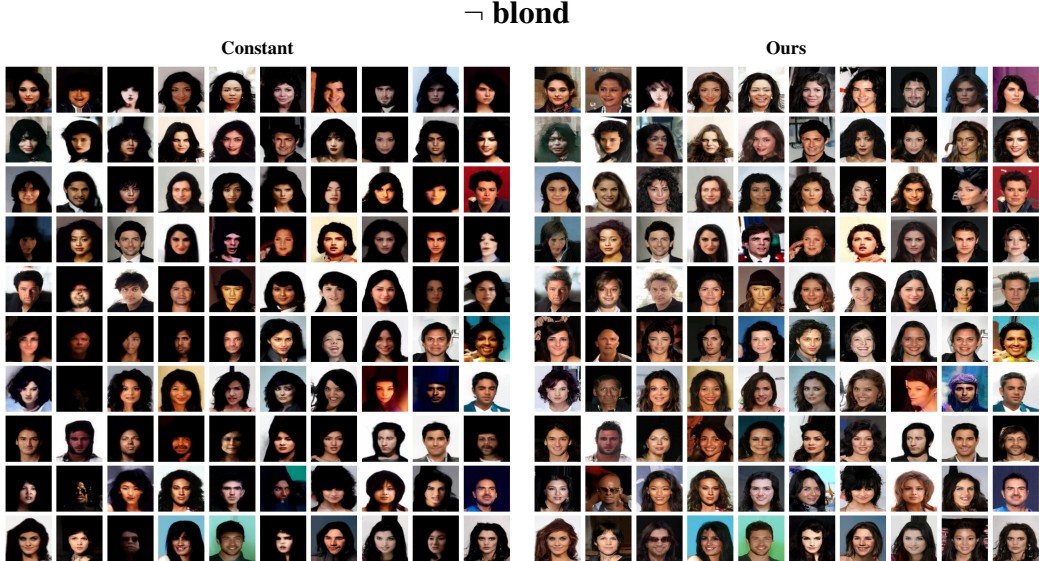

*Figure 13.* Batch of results for NOT composition (¬ blond) on CelebA. Conformity scores: Liu et al. (2022) = 1.0, LOGDIFF= 1.0.

