# OpenReview forum: "Logical Guidance for the Exact Composition of Diffusion Models"
_ICML.cc/2026/Conference — ICML 2026 regular_

### Official Review · Reviewer_sdcV · 2026-03-05

**Soundness:** 3
**Presentation:** 3
**Significance:** 4
**Originality:** 4
**Overall Recommendation:** 3
**Confidence:** 5

**Summary:**

The author proposed the LOGDIFF framework, which for the first time established a strict theoretical connection between Boolean logic and combinatorial guidance for diffusion models. This solved the core problem of existing methods, which relied on heuristic fixed weights and were unable to accurately handle complex Boolean logic expressions. The framework's effectiveness was verified in multimodal tasks such as image generation, protein structure, and drug molecule generation, providing a new logical guidance paradigm for the controllable generation of diffusion models.

**Compliance With Llm Reviewing Policy:**

Affirmed.

**Final Justification:**

I suggest keeping the score as it is.

**Key Questions For Authors:**

No Questions For Authors

**Limitations:**

yes

**Strengths And Weaknesses:**

Advantages:

1）For the first time, an exact calculus system was developed for the combination derivation of diffusion models based on atomic predicate Boolean formulas, and the sufficient conditions for the circuit representation of Boolean formulas were provided.
It bridges the gap between classifier-guided and classifier-free guidance, combines the atomic guidance score with the posterior probability estimation, separates the direction and weight of the guidance, and does not require obtaining gradients through backward propagation of the classifier.

2）Compared with the existing fixed-weight combination guidance methods, LOGDIFF employs dynamic coefficients determined by posterior probabilities, which solves the problems of pattern collapse and attribute mixing that are prone to occur in disjunction and negation operations with fixed weights.


Disadvantages:


1) In the experiment, the logical guidance effect was mainly verified using synthetic datasets (CMNIST, Shapes3D). Real-world images were only based on CelebA and ImageNet. For drug molecule generation, only the GRM5-RRM1 single protein pair was tested, and the generalization ability of the framework was not verified in more diverse and complex real-world scenarios.
2) The baseline comparison mainly focuses on the heuristic combination method with fixed weights. It only compares with methods such as Skreta et al. (2025b) and Du et al. (2023) in a small number of single-operation queries, but does not conduct a comprehensive comparison with the latest diffusion model logic-guided combination method CoInd.


3) In the dual-target drug design experiment, only 32 ligands were generated for each logical constraint and 8 replicate experiments were conducted. The small sample size would easily lead to statistical randomness in the results.

---

> ### Author Rebuttal · Authors · 2026-03-31
>
> We thank reviewer sdcV for the constructive feedback.
>
> # W1 & W3: Scope of Experiments
> Our experimental design aims to evaluate (i) logical composition complexity, (ii) generation quality, and (iii) generality across modalities.
> To this end, we validate complex queries (up to 5 AND/OR operations in Table 2) on datasets with large combinatorial feature spaces (e.g., Shapes3D with $4.8 \times 10^5$ possible combinations), where correctness of the Boolean expression can be evaluated precisely. We further assess generation quality on real-world datasets (CelebA, ImageNet), and demonstrate applicability to a different modality via molecular generation.
> While we presented the results with the same target as the previous work, we run additional experiments with 8 additional target pairs, batch size of 32 and 3 runs.
>
> | Method | (A * B) (↑) | A (↓) | B (↓) | Div. (↑) | Val. & Uniq. (↑) | Qual. (↑) |
> | --- | ---- | ---- | --- | ---- | ---- | --- |
> | TargetDiff | 64.69±7.20​ | −8.97±0.21​ | −7.17±0.76​ | 0.88±0.01​ | 0.94±0.04​ | 0.28±0.04​ |
> | DualDiff A∧B | 76.56±8.78​ | −8.88±0.31​ | −8.56±0.71​ | 0.89±0.01​ | 0.96±0.03​ | 0.27±0.09​ |
> | SDE A∧B | 76.50±7.76​ | −8.91±0.29​ | −8.53±0.61​ | 0.89±0.01​ | 0.98±0.01​ | 0.24±0.09​ |
> | LogDIFF A∧B | 77.18±7.96​ | −8.89±0.25​ | −8.63±0.67​ | 0.89±0.01​ | 0.99±0.01​ | 0.18±0.06​ |
> | LogDIFF A∨B | 75.67±8.02​ | −8.69±0.53​ | −8.63±0.68​ | 0.89±0.01​ | 0.99±0.01​ | 0.18±0.05​ |
>
> | Method | "Δ(A ,  B) (↑)" | A | B | Div. (↑) | Val. & Uniq. (↑) | Qual. (↑) |
> | --- | ---- | ---- | --- | ---- | ---- | --- |
> | DualDiff A∧¬B | 0.51±0.07​ | −13.27±0.21​ | −13.77±0.15​ | 0.86±0.01​ | 0.66±0.03​ | 0.00±0.00​ |
> | SDE A∧¬B | 0.67±0.30​ | −8.86±0.30​ | −8.49±0.72​ | 0.89±0.01​ | 1.00±0.01​ | 0.22±0.11​ |
> | LogDIFF A∧¬B | 0.66±0.26​ | −8.88±0.41​ | −8.57±0.74​ | 0.89±0.01​ | 0.97±0.04​ | 0.22±0.10​ |
> | LogDIFF A⊕B | 0.65±0.26​ | −8.89±0.26​ | −8.62±0.66​ | 0.89±0.01​ | 0.99±0.01​ | 0.17±0.06​ |
>
> Update and complete experiments on the additional 14 target pairs  will be added in the final version.
>
> # W2: Baselines
> We compare (in Appendix H.1) to the constant baselines as well as Skreta et al. (2025b), Liu et al. (2022) and Du et al. (2023) on the operations they define. These methods do not provide a complete Boolean framework (i.e., not all operators or recursive compositions are defined), which limits their applicability to multi-operator queries.
> We did not include Gaudi et al. (2025) as a direct baseline, as their work focuses on enforcing conditional independence during training rather than defining a general composition framework. Their examples correspond to constant weighting (e.g., $p(\neg(\neg C_1 \land \neg C_2)) \approx p(C_1)p(C_2)$), which is similar to our constant baseline. Their method is orthogonal to ours and we encourage to use it together with our framework, especially in the case of non-uniform datasets.
>
> We would like to thank the reviewer again for the feedback and hope that we have addressed their concerns. We are happy to answer any further questions.
>
>
> # Additional information on previous works
>
> We thank the reviewer for acknowledging the novelty of our work. We provide additional information with other recent frameworks, for other reviewers’ reference.
>
> We acknowledge that multiple alternative methods exist for deriving Boolean combinations of diffusion models.  The distinguishing feature of our approach is that we show that a consistent set of compositional rules can be given for arbitrary Boolean formulas, which follow directly from an event-based semantics dependent on the properties satisfied at terminal time, under certain sets of conditions (CI and/or ME). This is in contrast to approaches such as Blohm \& Garg, which apply Boolean operators at the distribution level; although such approaches may result in a calculus of distributions that satisfies the Boolean lattice axioms (using the max/min t-norms), or subsets of Boolean identities (such as De Morgan's laws), their probabilistic semantics are unclear.  This may have advantages; e.g. CI and ME assumptions not required. However, we would argue that the probabilistic relation to the original atomic properties is, in general, indirect, and CI and/or ME conditions remain relevant when a direct probabilistic semantics is desired. Part of our work thus involves characterizing the sets of conditions under which a consistent calculus (for which we use probabilistic circuits) can be given with an event-based semantics. As mentioned above, such conditions can rarely be fully guaranteed. The problem structure, however, may promote certain properties, such as ME between classes in a categorical variable. Others, such as CI, can be enforced during training by using the Gaudi et al. 2025 (CoInd) loss, or be taken to hold approximately.  When such conditions cannot be assumed, it is thus unclear whether a distribution- or event-based calculus is intrinsically preferred.

---

> > ### Author Rebuttal · Reviewer_sdcV · 2026-04-04
> >
> > Thanks to the authors' rebuttal and additional experimental results, my concerns have been resolved.

---

> > > ### Author Response · Authors · 2026-04-06
> > >
> > > We thank Reviewer sdcV for reading and acknowledging our responses. We would be grateful if the reviewer would consider increasing the review recommendation score.

---

### Official Review · Reviewer_Y1GF · 2026-03-08

**Soundness:** 3
**Presentation:** 3
**Significance:** 2
**Originality:** 2
**Overall Recommendation:** 4
**Confidence:** 2

**Summary:**

This paper proposed LOGDIFF, a variant of CFG that focuses on satisfying boolean logical constraints for diffusion models, such as AND, OR, and NOT. The core idea is to involve recursive composition rules for both posterior probabilities and guidance scores, such that complex logical formulas can be handled by combining atomic conditional scores from a single conditional diffusion model. The method is theoretically justified under assumptions such as conditional independence or mutual exclusivity. Experiments on multiple datasets show improved logical conformity scores over naive constant-weight guidance baselines, especially for OR and more complex formulas.

**Compliance With Llm Reviewing Policy:**

Affirmed.

**Final Justification:**

I increased my score to 4. The authors addressed most of concerns. My main concern remaining is the positioning of this work relative to prior works, raised by another reviewer. However, I do not feel confident in assessing whether this has been addressed by the authors. So I lowered my confidence score.

**Key Questions For Authors:**

- The method assumes access to atomic conditional scores p_t(x| c_i) for individual predicates c_i​ from a single conditional diffusion model. How exactly are individual predicates isolated at inference time? Does this require masking / null-tokenization of other attributes, separate embeddings, or additional training design?

- Could the authors better justify the practical importance of OR queries in the considered generation settings, and explain when the proposed OR-specific composition is meaningfully preferable to simply sampling one acceptable branch and using standard conditional generation?

- Similarly, for the NOT operator, could the authors clarify what realistic scenarios genuinely require the proposed posterior-based negation calculus, beyond common binary-attribute cases where negation might be handled by directly flipping the label?

- Could the authors provide additional qualitative examples to better demonstrate how the method behaves on more diverse and challenging logical compositions?

See weakness for other questions.

**Limitations:**

yes

**Strengths And Weaknesses:**

Strengths:

- The paper studies an interesting problem, i.e. how to involve Boolean logical constraints such as AND, OR and NOT in diffusion generation, and presents a recursive framework for doing so.

- The posterior-based formulation is more principled than simple constant-weight score mixing. Especially the formulation for OR and NOT.

-The proposed method is an inference-time plug-in and can be applied to existing conditional diffusion model.

Weakness:

-The novelty seems somehow limited, since prior work like CoInD, Gaudi et al, 2025 already considered Boolean score composition for AND/ NOT/ OR style operations. Therefore, the main advance here seems to be a more formal posterior-based refinement rather than introducing logical composition from scratch. The paper would benefit from a clearer positioning of what is truly new here beyond a more formal posterior-based refinement.

- Although the OR is an interesting boolean operator, the practical importance of OR remains a bit unclear. In many conditional generation settings, conjunctions and negations already seem to cover most use cases. So it will be helpful to have a stronger motivation for when OR-based constraints are genuinely necessary. If the practical goal of an OR query is just to obtain samples that satisfy “c1 or c2”, then one can instead sample c1 or c2 at random and then run standard conditional generation, which raises the question whether OR is actually necessary.

- The same for NOT operator. In common binary attribute settings, where negation may often be implemented simply by conditioning on the opposite label (e.g. using ‘not smiling’ directly rather than deriving a negation score from ‘smiling’). The paper would benefit  from clarifying realistic scenarios where these logical operators would be necessary.

- A key assumption of the method is the access to atomic conditional scores p_t(x | c_i), but the paper does not clearly explain how such atom-specific scores are obtained from a shared backbone in general multi-class conditioning settings.

- Since the method is purely an inference-time guidance scheme, it does not address whether the underlying diffusion model itself learns semantically reliable or disentangled atomic conditionals, which may limit robustness when the base model is misspecified.

---

> ### Author Rebuttal · Authors · 2026-03-31
>
> We would like to thank reviewer Y1GF for their thoughtful and constructive review. We address the reviewer's concerns below.
>
> The main novelty of our approach is in providing a consistent set of compositional rules for arbitrary Boolean formulas using an event-based semantics of properties satisfied at terminal time, under certain sets of conditions (CI and/or ME). While certain rules have similarities with prior compositional rules, our rule for NOT, the splitting of the OR rule into CI and ME, and the use of probabilistic circuits is non-standard. See Reviewer sdcV reply.
>
> # Q1/ W4: Conditional Score
> We train on a single feature while masking all others (e.g., “1, None” for a blue 1 in CMNIST). Accordingly, at inference time, we can access individual conditions via null-tokenization of all other attributes.
> We acknowledge that methods such as CoInd (Gaudi et al. 2025), which enforce conditional independence, can be integrated into our framework as plug-in loss.
> To isolate the effects of our composition rules and to ensure conditional independence, we adopt single-feature training on a uniform dataset. In this setting, we observed that CoInd did not improve the results.
>
>
> # Q2/ W2: OR-Composition
>
> Our OR-composition allows querying disjunctions that follow the learned data distribution.  For example, in the query $(1 \lor (2 \land \text{red}))$, the digit “1” is more likely than a red “2”, since all colors are allowed for “1”.
> This is reflected in the generated samples. In contrast to randomly selecting a condition (where accessing correct condition probabilities may be difficult), OR-composition preserves the learned distribution.
> Further, the combination rules generalize to disjunctions of arbitrary CI and ME propositions, which themselves may be complex Boolean formulas,  whose underlying probabilities are non-trivial to evaluate.
>
> # Q3/ W3: NOT-composition
> Our framework requires only conditional diffusion models for the positive atomic properties; hence, simply flipping the labels of a class would require costly retraining a new diffusion model conditioned on the negative labels. Further, the operator can be applied compositionally, so that we may derive the NOT model for a complex Boolean formula for which we have no positive labeled examples.
> NOT composition extends beyond binary settings. In the multi-class setting used in our experiments, it allows avoidance of specific outputs. This is also relevant in applications such as text-to-image generation, where certain attributes (e.g., nudity) should be avoided.  Additionally, the NOT operator is used in repulsive guidance to avoid confusions. In this case, the class most likely to be confused with the target is suppressed using NOT.
>
> # Q4: Qualitative examples
>
> We validate complex queries (up to 5 AND/OR operations) on datasets with large combinatorial feature spaces (Shapes3D $4.8 \times 10^5$ possible combinations), including qualitative results (CelebA, ImageNet) and multiple modalities (real and synthetic images, molecular generation). We will extend the CelebA experiments to include multi-class features for improved qualitative evaluation and further extend the molecular generation experiments.
>
> # W1: Novelty compared to CoInd
>
> We agree that Gaudi et al. address logical composition by enforcing conditional independence during training. This is complementary to our approach and can be integrated into our framework.
> However, their work does not provide a general, explicitly defined framework for  composition of arbitrary Boolean expressions. While they demonstrate specific compositions, extending these to more complex expressions is not formalized. From their examples, conjunction and negation correspond to constant weighting, and their disjunction is approximated as $p(\neg(\neg C_1 \land \neg C_2)) \approx p(C_1)p(C_2)$, which is similar to our constant baseline.
> In contrast, our method enables probability-dependent composition of arbitrarily complex Boolean expressions derived from an event-based semantics (see reviewer DDQV14 above) and provides explicit construction rules for combining scores. In particular, the derived operators (especially NOT and OR-CI), the use of probability circuits over sub-expressions, and the explicit connection to Boolean logic are novel contributions, to the best of our knowledge. We will clarify this distinction and further improve the positioning with respect to prior work.
>
>
> # W4: Misspecified model
>
> We agree that our method operates at inference time and does not address potential misspecification of the underlying diffusion model. Our framework is compatible with training-time methods such as CoInd (Gaudi et al., 2025) or other approaches that improve conditional disentanglement and robustness.
>
> Once again, we would like to kindly thank the reviewer for taking their time to review our paper and provide constructive feedback. We are happy to discuss any further questions or concerns in the next stage.

---

> > ### Author Rebuttal · Reviewer_Y1GF · 2026-04-03
> >
> > My original concerns were mostly addressed. However, after reading another reviewer's reviews. I am a bit concerned about the positioning of this paper with prior works. I will adjust my score after assessing if this concern has been properly addressed.

---

> > > ### Author Response · Authors · 2026-04-06
> > >
> > > We thank Reviewer Y1GF for reading and accepting our answers to the reviewer’s concerns.
> > >
> > > # Position of the paper
> > >
> > > One of Reviewer DDQV’s main concerns is related to the concurrent work of Blohm & Garg 2026 (accepted Jan 2026, concurrent to our submission), which extends the work of Garipov et al. 2023, which we cite.  We note that these works do not provide a general framework for Boolean compositionality of events at terminal time; instead, they provide a mechanisms for probability density function (pdf) composition, principally using products and mixtures of experts to implement AND and OR respectively combined with various forms of NOT operator.  Blohm & Garg 2026 proposes an alternative approximate Boolean framework for the composition of pdfs. The framework is exact only in the limit of $\lambda \to \infty$, but it provides an elegant way to combine fuzzy logic with probability distributions, and therefore the related score functions. Our work, however, proposes a way to implement exact score compositionality (subject to assumptions discussed in the response to reviewer DDQV) respecting all Boolean identities, in particular the De-Morgan laws and associativity. We also provide a practical algorithm to derive the composition coefficients, and associated theoretical results, utilizing the bridge we develop between score composition and probabilistic circuits.
> > >
> > > A second major concern from Reviewer DDQV, is the relationship between the operators used in our derivation and the ones previously proposed. We address this relationship in the response to Reviewer DDQV.
> > >
> > > We will further improve the positioning with respect to prior works by making the following alterations in the text:
> > >
> > > Line 027: “As a result, there is no formal framework for translating complex logical expressions [...]” will be changed to “In concurrent work, (Blohm and Garg 2026) provide a fuzzy-logic-based framework for composing distributions; while this work allows complex Boolean expressions to be represented, Boolean identities are only approximately satisfied, and the framework does not directly ensure compositionality of properties at terminal time.”
> > >
> > > Line 420:  “replacing fixed weights with probability-dependent posterior coefficients.” will be changed to “drawing on related work (Li et al., 2023; Skreta et al., 2025b) to replace fixed weights with probability-dependent posterior coefficients”.
> > >
> > > Line 422: “A parallel line of work studies model composition, combining separate pre-trained models to merge their capabilities.” will be changed to “A parallel line of work studies Boolean compositions of density functions, for example derived from pre-trained models, to merge their capabilities.”
> > >
> > > We thank the reviewer for the effort in reading and considering our responses and we would be thankful if the reviewer would increase the recommendation score.

---

### Official Review · Reviewer_N4Be · 2026-03-12

**Soundness:** 4
**Presentation:** 3
**Significance:** 4
**Originality:** 3
**Overall Recommendation:** 5
**Confidence:** 4

**Summary:**

A novel approach to enrich diffusion models with logical guidance in a
an efficient and principled way.

**Compliance With Llm Reviewing Policy:**

Affirmed.

**Final Justification:**

A relevant contribution to the field of constrained generative AI. This is a very timely topic, the proposed solution is sound and effective and it has the potential to foster further work in the area.

**Key Questions For Authors:**

none

**Limitations:**

yes

**Strengths And Weaknesses:**

Constrained guidance is a crucial aspect to enable diffusion processes
to generate samples satisfying formal requirements. This approach
introduces a principled approach to incorporate complex logical
guidance in the inference stage of diffusion models, overcoming the
limitations of existing solutions that are either heuristic or not
general.

The proposed formulation is sound and adapts in a non-trivial way
 findings from the literature on probabilistic circuits.

The experimental evaluation covers multiple settings, from synthetic
datasets to high resolution images to molecules and proteins.

---

> ### Author Rebuttal · Authors · 2026-03-31
>
> We thank Reviewer N4Be for the positive and thoughtful assessment of our work. We are happy to read that the reviewer highlights the principled nature of our approach and recognizes the connection to work on probabilistic circuits. We also appreciate the acknowledgment of our experimental evaluation across diverse settings and modalities.

---

> > ### Author Rebuttal · Reviewer_N4Be · 2026-04-05
> >
> > My original evaluation was already positive and I didn't have  major concerns to raise.

---

### Official Review · Reviewer_DDQV · 2026-03-14

**Soundness:** 4
**Presentation:** 2
**Significance:** 3
**Originality:** 4
**Overall Recommendation:** 4
**Confidence:** 4

**Summary:**

The manuscript proposes a procedure for combining score signals from diffusion models with logical expressions and states the conditions under which these compositions are exact.
The operators proposed include the logical negation, conjunction (product of experts), CI-disjunction (probabilistic sum), and ME-disjunction (mixture of experts).
Based on a normalized probability estimate $\pi$, they define recursive rules for aggregating scores and posteriors via scalar reweighting.
This results in linear combinations of base-scores in all cases.
Further, as a separate small contribution, they propose repulsive guidance, where the sample is repelled from similar classes to the target class to improve quality and avoid class "confusion".

**Compliance With Llm Reviewing Policy:**

Affirmed.

**Final Justification:**

The theoretical contribution of the work seems solid and principled.

However, the manuscript still does not sufficiently position itself with respect to prior work, which, in my view, weakens the overall contribution.
The inclusion of a fairer baseline, such as Skreta’s “or,” and the discussion of similarities between the proposed operators and prior work from preceding years, such as Garipov et al. and Skreta et al., already improve this aspect.
Nevertheless, I believe the manuscript would be positioned much more strongly if it clarified more explicitly how the posterior-likelihood formalism relates to other methods based on density estimators or classifier guidance.
At present, the discussion of prior work comes across as somewhat defensive rather than comparative.
For example, in their latest reply, the characterization of an arXiv manuscript (https://arxiv.org/abs/2410.14398 according to arXiv first submitted in 2024) as “concurrent” is, in my view, an example of that tendency.

That said, I still think the contribution in the manuscript is worthy of acceptance, and I will increase my evaluation to 4.

**Key Questions For Authors:**

**Q1 (Ad. W4): Design Choices for $\pi$ in experiments and related costs/constraints.**

Based on the manuscript, it is not quite clear to me how $\pi$ is estimated in the experiments and in practice.
- Could the authors please elaborate on their different choices and how precisely $\hat\pi$ is constructed from, e.g., a density estimator? What assumptions about the model need to hold to use the respective method? How much does each method impact the inference time?
- For noise-aware classifiers, how much overhead do these classifiers incur? As the diffusion process is dynamic, the behavior of the composition *over time* may be very sensitive, e.g., because score magnitude changes and different time-steps/noise levels are associated with different semantic features.
  Do noise-aware classifiers struggle in high-noise time? Would the authors expect/or did they observe that controlling the prediction confidence of a classifier changes the composition behaviour?

**Q2(Ad. W3) CI/ME Conditions**

**Q2A: Why two operators?** The manuscript proposes two different disjunction operators, but effectively, if two concepts are mutually exclusive, then $\pi_{1}\pi_{2}\ approx 0$, so should these two forms of disjunction not collapse in principle?
If so, how will using one operator over the other change the behaviour (intuitively) if it is unclear whether the concepts are ME?

**Q2B Checking the CI Assumption** The assumption of conditional independence seems very strong and counterintuitive to me.
In my understanding, this would not only need to hold over the final samples, but also over the noised samples, where semantics are very weird and not easily understood.
Is there any argument for why this should hold in practice? Do the authors check this to some degree?

**Q3 (Ad. W5) Experimental Details**
Could you please answer the following questions from **W5**:
- What is the meaning of a "not" prompt/formula? (especially with repulsive guidance)
- How precisely do all methods in Figure 3/Table 3 look mathematically? (completely, as plugged into the sampler) The results seem inconsistent (should the two baseline disjunctions not be the same?)
- Why does Table 3 not feature the ME results?

**Q4: Guidance Strength** I am not sure if I understand the atom aggregation rule correctly, but:
Assume I have a conjunction of three scores $s_1,s_2,s_3$ with the unconditional $s_\emptyset$.
Is the final score then $s_1+s_2+s_3-3s_\emptyset$?
Is this compounding unconditional score fine, or does it lead to issues if the effective guidance strength becomes too much?
Similarly, is a pure negation just $s_\emptyset-s_1$ scaled by some non-negative scalar?
As such, for low $\pi$ is this not just random noise, and for high $\pi$ divergent from $s_1$? This part is not quite clear to me.


**Note: Relevant Recent Work** I already mentioned it above, but there is a very similar paper by Blohm & Garg (2026) that is not cited.
I realize this is recent, but it is still relevant to this work, and the authors should check this out.
The objective of this manuscript and that paper seems to be exactly the same.

**Limitations:**

Yes

**Strengths And Weaknesses:**

I like the paper's contribution a lot and personally think that the compositionality of diffusion models is extremely useful.
The contribution is theoretically well-grounded and seems very sound and thorough in scope.
This also holds for experimental evaluation, which features a wide array of settings.

However, the manuscript positions itself poorly with respect to related work, claiming to be the first paper that provides a solution for composing diffusion models with complex formulas, which is inaccurate (**W2**).
Further, very similar **approaches and contributions in closely related work are not properly recognized**. This seems especially strange considering that most experiments are clearly adapted from other papers.

In the experiments section, the **the manuscript chooses baselines that are completely unreasonable (disjunction and maybe negation, see W1).**
Finally, the paper does not fully explore practical questions (e.g., how actually to obtain $\pi$) and assumptions (e.g., whether the concepts are actually CI).

## Verdict

**In Summary, I feel I have to score this manuscript with a weak reject, mostly due to a narrative that does not properly recognize related work and establishes unrealistic state-of-the-art assumptions.
I think the manuscript needs to be rewritten to be fully intellectually honest.**
Aside from this issue, I believe the CI assumption remains a gap between theory and practice.
If the authors have some way to test whether/quantify how much this holds in practice, the theoretical basis would be extremely strong.

## Strengths

**S1: Strong theoretical basis for the operator set**

Composition is tackled by many papers, but the line between theoretically well-motivated and heuristic methods blurs.
This approach seems very sound from a theoretical perspective, and ties in neatly to the Product of Experts/Mixture of Experts interpretation.
Negation often is a tricky operator to implement,

**S2: Thorough Experimental Evaluation and high-quality appendix.**

The paper provides substantial supporting work in the appendix, including studies of discrete settings, further characterizations of relevant distributions, etc.
Similarly, the manuscript replicates many experiments from related work and, in the appendix, even compares different methods for estimating posterior probabilities.
I think both these aspects improve the quality of the manuscript and make it more impactful.

**S3: Repulsive Guidance is a bit orthogonal to the main contribution, but a very nice technique**

The repulsive guidance approach reminds me of the paper by Kirchhof et al. (2024). It seems very useful not only for improving generation quality but also as a more general "guardrailing" technique to avoid certain outputs.

The principal idea of avoiding some output is not groundbreaking. Still, it very nicely highlights what you can do with more complex composition techniques, and adaptively avoiding similar classes to avoid confusion is a very neat variation of this.


## Weaknesses


**W1: Unreasonable Baseline Assumption for disjunction (and maybe negation)**

**W1A:** A repeated statement in the manuscript is that disjunctions are apparently modelled by score averaging.
Consequently, the authors claim that the geometric mean $(p_1(x)p_2(x))^{1/2}$ is a reasonable baseline for disjunction when the product $(p_1(x)p_2(x))$ is a conjunction.
I am **strongly opposed to this baseline assumption for multiple reasons**:
- It makes no sense (to me) from a probabilistic interpretation.
- Product and geometric mean are often treated interchangeably as a conjunction. The manuscript explicitly states, at one point in the introduction, that the geometric mean is a conjunction.
- I have never seen this interpretation in any work that tries to model logical operations on probability distributions. If there is work that explicitly models a disjunction like this, please show.
- The manuscript cites multiple papers that explicitly find other ways to avoid this completely unfounded baseline.
  Constraining related work to **diffusion** that would be, for example, Skreta et. al. (2025a), Blohm & Garg (2025,2026), and Garipov et. al. (2023) (when given access to an unconditional model that acts as logical "TRUE", then their contrast operator is functionally complete).
This, in my opinion, invalidates the disjunction baseline, especially in Figure 3 and Table 3.
**Further, the OR operator by Skreta et. al. (2025a) is equivalent to the ME disjunction in this manuscript.**

**W2b:** The baseline negation as written near line 267 is also ambiguous.
Based on Figure 1 B, this seems to me like the EBM style negation, $s_\emptyset - s_c$; which is famously unstable (see, e.g., Garipov et. al. (2023) for an argument about this).
A far more commonly used form of "negation" is precisely the CFG form with some parameter $\gamma$ where "not c" is modelled $s_\emptyset + \gamma(s_c - s_\emptyset)$ (e.g., Ho et. al. 2022, Skreta et. al. 2025b).
This is much more commonly used and much more stable in practice.
The crucial factor here is that the "coefficients" of the score sum to 1, so the guidance does not degenerate when $s_c\approx s_\emptyset$.

**W2: Overclaiming of contributions with respect to related work.**

"There is no formal framework for translating complex logical expressions into corresponding guidance dynamics for diffusion models." (Introduction near line 27) is a very strong and inaccurate claim.
This feels like it discredits the work of
- (cited) Garipov et. al. 2023: "Compositional sculpting of iterative generative processes" which performs composition (also of diffusion models) with non-standard but logically expressible operators and *noise classifiers*. (Their contrast operator seems to be functionally complete.)
- (cited) Gaudi et. al. 2025: "Coind: Enabling logical compositions in diffusion models" that tackle logical composition of diffusion models with classical connectives and propose a training procedure to improve the composition of *arbitrary formulas*.
- (cited) Skreta et. al. 2025: "Superposition of Diffusion models" that do not seem to model negation, but have "and" and "or" operators with *online density estimation.*
- (not cited) Blohm & Garg 2026: "Composition of Pretrained Diffusion Models: A Logic-Based Calculus" that proposes operators to model logical formulas similar to this manuscript, and uses *online density estimation*.
This ignores many other papers that investigate similar directions, e.g., for EBMs.

This manuscript also claims in the related work that the objective of *model composition* differs from that of "composition of diffusion models" (?) in the presented manuscript, because *model composition* "combines capabilities of different models".

**In summary: There are multiple papers with the same/very similar objective, and many use essentially the same trick as $\pi$ in the manuscript: get some notion of likelihood to perform operations that require dynamic weighting.**


**W3: Validity of CI/ME assumption in typical diffusion models not established.**
I think it is a very nice theoretical contribution to define *when* score composition works precisely.
However, I fear that the neat definition of *categorical-CI-queries* becomes very messy, e.g., in prompt-based systems and with noisy samples.
It is not obvious that intuitive semantic characteristics should translate mathematically into noisy diffusion models, and the manuscript did not explicitly establish or test this.

**W4: posterior $\pi$ with a noise classifier requires expensive training, and the estimators are very noisy. Details in the experiments are not specified.**

The posterior probability (estimator) $\hat\pi$ is, as mentioned in **W2**, very similar to approaches by multiple prior methods.
The manuscript vaguely mentions noise-aware classifiers and, in the appendix, also compares them to the Itô density estimator.
However, different choices for $\hat\pi$ seem applicable only in different settings and incur different costs.
- Noise-aware classifiers are very costly to train.
- Density estimators need extra work (and assumptions) to estimate the posterior $\pi$.
- The zero-shot classifier approach by Li et al. (2022) seems to be very noisy and, based on the formulation in this manuscript, relies on enumerating over concepts $c_i$, which is a constraint on the model.

**(Minor) W5: Some experimental results unclear or misleading.**
- Figure 3:
	- Disjunction *baselines* are different for ME/CI, how can this happen, i thought this was the same in both cases?
	- Low Disjunction Entropy consistent with this operator essentially being a conjunction.
	- What is the intended output for a "not" prompt?
- Table 3:
	- Again the meaning of an isolated "not" is unclear here, what even is the target?
	- Besides "not," this experiment does not show strong improvements using the proposed method.
	- Why is there no OR-ME here? IIRC, multiple hair colors could be used here.
- Table 4:
	- What is the meaning of "not" *with repulsive guidance*?
- Why is FKC now used in the experiments? The difference in the FKC between tables 7 and 8 and the base variants appears to be a side effect of SMC collapsing the particle population (diversity decreases). What FKC terms are even used for these operators? If I understand the work of Skreta et al. (2025b) correctly, FKC depends on partial differentials over time; do they depend on the behaviour of $\pi$ too?

I think the experiment section would be easier to follow (and more convincing) with a more focused set of results, clearly explained and motivated, with the relevant details spelled out.


### Used Sources

Ho, Jonathan, and Tim Salimans. "Classifier-free diffusion guidance." arXiv preprint arXiv:2207.12598 (2022).
Garipov, Timur, et al. "Compositional sculpting of iterative generative processes." Advances in neural information processing systems 36 (2023): 12665-12702.
Li, Alexander C., et al. "Your diffusion model is secretly a zero-shot classifier." Proceedings of the IEEE/CVF International Conference on Computer Vision. 2023.
Kirchhof, Michael, et al. "Shielded Diffusion: Generating Novel and Diverse Images using Sparse Repellency." arXiv preprint arXiv:2410.06025 (2024).
Skreta, Marta, et al. "The Superposition of Diffusion Models Using the It\^ o Density Estimator." arXiv preprint arXiv:2412.17762 (2025a).
Skreta, Marta, et al. "Feynman-kac correctors in diffusion: Annealing, guidance, and product of experts." arXiv preprint arXiv:2503.02819 (2025b).
Blohm, Peter, and Vikas K. Garg. "Hot Fuzz: Temperature-Tunable Composition of Diffusion models with Fuzzy Logic." The Fourteenth International Conference on Learning Representations.

---

> ### Author Rebuttal · Authors · 2026-03-31
>
> We thank the reviewer for the very detailed and insightful review. We highly appreciate the effort invested in providing such thorough feedback.
> # Q1/W4 Design Choices
> We estimate $\hat{π}$ using a classifier, in CMNIST, x2 runtime. Noise-aware classifier results in (\~5x for CelebA) training. We will include classifier training details and computation time comparison.  For Skreta et al., we use their likelihood estimator. In our ODE setting, likelihood updates require a divergence computation, which takes \~4x longer, but is shorter in the SDE setting.  At high noise levels, predictions become near-uniform, leading to equal weighting, which does not harm composition. While estimator bias could affect composition, this was not observed. Our framework is estimator-agnostic.
> # Q2/ W3 CI/ME
> **Q2A** The two operators for OR are derived with different assumptions, one based on CI and the other ME. It is possible to transform the OR-ME into the OR-CI rule by setting $\pi(\chi)\pi(\phi)=0$. However, this is not a valid derivation, since the properties $\chi$ and $\phi$ cease to be CI if they are ME; rather this reflects their derivation from the general OR decomposition on line 580.  However, the general form not useful because it would require the knowledge of joined probability $\pi(\phi\wedge\chi)$.
> **Q2B** The conditional independence is on the terminal event given the current sample. Our framework could be extended to high-order statistics but at expense of complexity. We note also that CI is a commonly assumed in previous works. We can use Gaudi et al. enforce conditional independence during training as plug-in loss. However, they do not provide a general framework for recursive Boolean expressions and their examples coincide for AND and NOT with the constant baseline.
> # Q3/W5
> The NOT output consists of all configurations except the negated one (e.g., ¬9 generates number 1-8).
> **Fig 3**: We sum unconditional and logical scores (Alg. 1). For illustration, consider the query $¬3$. Our compositional score is $s_∅+w⋅-\frac{\hat{π}(3)}{1-\hat{π}(3)}(s_3-s_∅)$ vs. the constant baseline: $s_∅+w⋅-(s_3-s_∅)$. For the disjunctions, the constant baseline is identical, but behavior differs: in CI-OR (e.g., 3 ∨ red), large guidance weights lead to conjunction-like collapse, which is not the case for ME-OR (e.g., 1 ∨ 2), as visible in the qualitative results (App. H).
>
> **Repulsive guidance** avoids confusing classes by extending the atom $c$ to $c∧¬c_{not}$. Combined with negation, this induces a double-negation effect: samples are pushed away from the negated class and attracted to the most likely alternative. The resulting scores are
> $s_∅+-\frac{\hat{π}(c)}{1-\hat{π}(c)}((s_c-s_∅)+-\frac{\hat{π}(c_{not})}{1-\hat{π}(c_{not})}(s_{c_{not}}-s_∅))=s_∅-\frac{\hat{π}(c)}{1-\hat{π}(c)}(s_c-s_∅)+\frac{\hat{π}(c)}{1-\hat{π}(c)}\frac{\hat{π}(c_{not})}{1-\hat{π}(c_{not})}(s_{c_{not}}-s_∅).$
> We will add an example in App. B.
> For **Tab. 3** CelebA, the goal is to show that composition does not degrade quality and results are similar to the baselines. Following Gaudi et al., we use two binary attributes, allowing only CI-OR. We will extend this to multi-class attributes (hair colors) to evaluate ME-OR.
> # Q4: Guidance
> We add unconditional score to the logical (Alg. 1). For conjunctions, this yields $s_∅ + \sum_i (s_i - s_∅)$. For negation: $s_∅+w_{not}⋅-(s_i-s_∅)$. The not-weight approaches zero for unlikely negated classes and becomes strongly negative otherwise, effectively repelling them (clamped in practice App. E). So the coefficients always sum to 1, and we observe stable behavior even for complex compositions (Tab. 4). We will clarify this further in Alg. 1 and our example (App. B).
> # W2: Related Work (see also Reviewer sdcV additional info)
> We propose a score-based framework for composing logical expressions using probability-weighted scores. Prior work addresses related problems but differs in formulation.
> Garipov et al. introduce conjunction and a contrast operator, but logical composition is not derived or validated. The method is not score-based and relies on classifier gradients.
> For discussion on Gaudi et al. see W1 Reviewer Y1GF below.
> Skreta et al. define ∧/∨ with a density estimator. Their OR matches our OR-ME, while AND requires solving a linear system and does not trivially extend to recursive compositions. NOT is not defined. Concurrent work (Jan 2026) by Blohm & Garg similarly proposes logical recursive composition rules, but uses a λ-parameterized fuzzy calculus and FKC, resulting in different composition rules. We will revise the paper to better position our method relative to these works.
> # W1: Baselines
> We will move additional baselines (Skreta et al., Liu et al.) from the App. H to the main text, and compare to Garipov et al. and Blohm & Garg as soon as their code is available. We agree that the OR baseline (line 269) is non-standard; we will refer to it as a 'constant baseline' rather than Disjunction.

---

> > ### Author Rebuttal · Reviewer_DDQV · 2026-04-04
> >
> > I thank the authors for their effort in the rebuttal. I have raised many points, and the limited space in the rebuttal makes it difficult to address them all. I appreciate the effort to fit everything.
> >
> > **Q1/W4**: Thank you for the clarification. I believe it is important to mention these aspects in the main manuscript.
> > I believe the paper's estimation-agnostic stance is a nice feature, but obtaining well-behaved estimators remains nontrivial and should be discussed.
> >
> > **Q2/W3**, **Q2A** has been resolved, you are right.
> > For **Q2B**: thank you for the explanation, but I wonder: if the assumption holds only terminally, how is it valid to apply the operator to intermediary distributions? Would the same assumptions not need to hold at any given point in time to apply the operators for the intermediary distributions?
> > E.g., at early stages of the process, I expect the scores to be highly aligned, regardless of whether they are CI/ME later.
> >
> > **Q3**
> > The baseline *not* operation seems reasonable.
> >  > For the disjunctions, the constant baseline is identical, but behavior differs.
> > So, if I understand correctly, Fig 3 shows different experimental setups for the ORs. This was slightly confusing because it wasn't mentioned in the figure caption.
> >
> > **Q4** This seems mostly fair. For this discussion, my question has been addressed, but I still wonder whether using base scores with a negative coefficient for the disjunction of three scores might lead to potential issues.
> >
> > **W1** is not resolved by adding baselines. **The constant weight *or* baseline is misleading and should be removed**. It implements an operator that is commonly interpreted as a conjunction, because it is the same operation, with smaller coefficients for the scores.
> >
> > For **W2**, I still feel the authors do not clearly state that many of the works they mention produce very similar operators and, in particular, rely on some form of likelihood estimation.
> > I like the clear formalism for the posterior $pi$, but can you connect it more clearly to how prior work uses very similar mechanisms and how they map to each other?
> > Especially because mechanisms from e.g., Skreta are specifically reused in the experiments.
> >
> > ## Summary
> > In total, my *main questions have mostly been addressed*, and I am potentially willing to raise my score. I remain somewhat skeptical about the paper’s positioning relative to prior work and some of the baselines.
> >
> > In particular, I believe the paper would benefit from more clearly acknowledging that several of the operators and the posterior likelihood mechanism are closely connected to existing ideas, and that one of the paper's contributions is to unify and streamline these scattered results. At the same time, the or-ci operator does appear genuinely new and particularly interesting, and the clean formalism for the posterior likelihood is also a strength that could be emphasized more clearly.
> > The constant-weight baseline should not, in any capacity, be conflated with a disjunction, in my opinion.
> >
> > I would appreciate it if the authors could briefly highlight here, in particular, the most important *similarities* to prior work and the respective operators there.

---

> > > ### Author Response · Authors · 2026-04-06
> > >
> > > We thank the reviewer for their detailed comments on our work, and give our responses below.
> > >
> > > # Q2: CI/ME assumptions at non-terminal time
> > >
> > > Comment: Is it valid to apply the operators to intermediary distributions?
> > >
> > > Response: In our theoretical analysis, we analyze the case that each AND node in a probabilistic circuit is associated with properties that are CI conditioned on a particular $x_t$, and each OR node is associated with properties that are CI or ME conditioned on $x_t$. As analyzed in Prop. D.1 and Prop. D.2 case (2) in the Appendix, properties which are ME at terminal time are guaranteed to be ME when conditioned on $x_t$ for $t>0$, and hence our model is exact for properties which can be expressed using combinations of NOT and OR-ME only. However, properties which are CI at terminal time are not guaranteed to be CI for all $x_t$. Our method thus makes the assumption that CI is a useful approximation in the particular cases we investigate. CI may be explicitly optimized for all intermediate distributions using an approach such as CoInd (Gaudi et al. 2025); as noted in our previous response to rev. Y1GF, however, we tested the CoInd training loss with our approach, and found no improvement in performance. This may suggest that the performance is affected by a trade-off between the degree to which CI holds for all $x_t$, and the consistency of the unconditional and conditional models with the data distribution. Given the prevalence of the CI assumption in related approaches, we believe this remains an important area of investigation for future work.
> > >
> > > # W2/Relationship to prior work
> > >
> > > Comment: In particular, I believe the paper would benefit from more clearly acknowledging that several of the operators and the posterior likelihood mechanism are closely connected to existing ideas, and that one of the paper's contributions is to unify and streamline these scattered results. At the same time, the or-ci operator does appear genuinely new and particularly interesting […] I would appreciate it if the authors could briefly highlight here, in particular, the most important similarities to prior work and the respective operators there.
> > >
> > > Response: We will add a paragraph based on the following:
> > >
> > > “We summarize the relationship of our operator set to prior and concurrent work as follows: (a) to our knowledge, our NOT operator has not been investigated in prior work, aside from work developed concurrently and independently of ours (https://arxiv.org/abs/2410.14398), which we were made aware of only after our manuscript submission, and which was proposed not as a compositional operator, but in the context of negative guidance; (b) our AND operator has been widely used, for instance in Gaudi et al. 2025, Skreta et al. 2025a, and Blohm and Garg 2026, although without explicitly stating a CI assumption except for Gaudi et al. 2025; (c) our OR-CI operator has not to our knowledge been considered in prior work; (d) our OR-ME has been widely used in previous works as disjunction, including Skreta et al. 2025b, and Blohm and Garg 2026, although without explicitly stating its dependence on an ME assumption.  Our approach allows these operators to be unified in a common framework, in which we can demonstrate their sufficiency given certain sets of assumptions on the distributions they are applied to.”
> > >
> > > Further, we will improve the positioning by making the text alterations noted in the response to rev. Y1GF below.
> > >
> > > # W1/Disjunction baseline
> > >
> > > Comment: W1 is not resolved by adding baselines. The constant weight or baseline is misleading and should be removed.
> > >
> > > Response: We will remove the constant baseline for the OR. Since the constant-weight disjunction baseline is problematic, we suggest (if the reviewer agrees) that: Firstly, in place of the ‘constant’ Disjunction baseline (line 269), we will use the OR operator from Skreta et al. 2025b (equivalent to our OR-ME), which, as noted above, is commonly used as OR in prior work, irrespective of CI or ME assumptions. Further, we will update Tables 2 and 3, and Figs. 2 and 3, to reflect this change. Hence, the OR-ME column on Tab. 2 and the OR-ME subplot on Fig. 3 will be removed (since we are using this as our OR baseline), and the OR-CI subplot in Fig. 3 will be adapted to use OR-ME as the baseline. Further, the ‘Constant’ model on Tab. 2 will be relabeled ‘Baseline (ablated)’, and this row will be updated using the baseline operators, with OR-ME uniformly used as the OR operator (alternatively, Tab. 2 may be removed in its entirety if recommended by the reviewer). We will further adapt Fig. 2, so that the visual examples are derived from the Baseline (ablated) model (removing the example (4 OR 1), which is an OR-ME combination). Finally, we will add the results from Tab. 11 in the main text, summarizing the additional baselines compared on this table.
> > >
> > > We thank the reviewer for their effort, and we would be grateful if the reviewer would consider increasing their score.

---

### Decision · Program_Chairs · 2026-04-30

**Decision:**

Accept (regular)

**Comment:**

This paper is an approach to complex logically constrained guidance of diffusion models. Overall reviewers liked the direction of the work, but the complaints centered on 2 things: the positioning of the paper wrt prior work both in terms of scholarship but also the resulting impact on novelty. Through the rebuttal the situation seems to have improved but it still detracts somewhat from the value. The other issue was how comprehensive the experiments were and whether it used appropriate baselines (mentioned multiple times). The reviewers seem to indicate they found the responses satisfactory though Reviewer sdcV seems leery of raising score to accept based on novelty.